# Synthetic virology approaches to improve the safety and efficacy of oncolytic virus therapies

Taha Azad [1,2,3,15], Reza Rezaei[1,4,15], Ragunath Singaravelu [1,4], Adrian Pelin[5], Stephen Boulton [1], Julia Petryk[1], Kemal Alper Onsu[1], Nikolas T. Martin[1], Victoria Hoskin [1], Mina Ghahremani[6], Marie Marotel [1,4,7], Ricardo Marius[1], Xiaohong He[1], Mathieu J. F. Crupi[1,4], Huy-Dung Hoang[4,8], Abolfazl Nik-Akhtar[4,9], Mahsa Ahmadi[10], Nika Kooshki Zamani[11], Ashkan Golshani[9], Tommy Alain [4,8], Peter Greer[12], Michele Ardolino[1,4,7], Bryan C. Dickinson [13], Lee-Hwa Tai[3,14], Carolina S. Ilkow[1,4] & John C. Bell [1,4] ✉

The large coding potential of vaccinia virus (VV) vectors is a defining feature. However, limited regulatory switches are available to control viral replication as well as timing and dosing of transgene expression in order to facilitate safe and efficacious payload delivery. Herein, we adapt drug-controlled gene switches to enable control of virally encoded transgene expression, including systems controlled by the FDA-approved rapamycin and doxycycline. Using ribosome profiling to characterize viral promoter strength, we rationally design fusions of the operator element of different drug-inducible systems with VV promoters to produce synthetic promoters yielding robust inducible expression with undetectable baseline levels. We also generate chimeric synthetic promoters facilitating additional regulatory layers for VV-encoded synthetic transgene networks. The switches are applied to enable inducible expression of fusogenic proteins, dose-controlled delivery of toxic cytokines, and chemical regulation of VV replication. This toolbox enables the precise modulation of transgene circuitry in VV-vectored oncolytic virus design.

Tumors are heterogeneous cellular ecosystems that continuously evolve in the face of therapeutic intervention limiting the efficacy of conventional treatments for metastatic disease. Replicating cancer therapeutics like engineered bacteria[1], immune cells[2] and viruses[3] comprise an emerging class of modalities that, in principle, can work in harmony with a patient's immune system to initiate robust, systemic anti-tumor immune responses capable of keeping pace with and attacking evolving tumor ecosystems. These forms of therapies can be

[1]Ottawa Hospital Research Institute, Ottawa, ON K1H 8L6, Canada. [2]Faculty of Medicine and Health Sciences, Department of Microbiology and Infectious Diseases, Université de Sherbrooke, Sherbrooke, QC J1E 4K8, Canada. [3]Centre de Recherche du CHUS, Sherbrooke, QC J1H 5N4, Canada. [4]Department of Biochemistry, Microbiology and Immunology, University of Ottawa, Ottawa, ON K1H 8M5, Canada. [5]Quantitative Biosciences Institute (QBI), University of California San Francisco, San Francisco, CA CA 94158, USA. [6]Department of Biology, University of Ottawa, Ottawa, ON K1N 6N5, Canada. [7]Center for Infection, Immunity, and Inflammation, University of Ottawa, Ottawa, ON K1H, Canada. [8]Children's Hospital of Eastern Ontario Research Institute, Ottawa, ON K1H 8L1, Canada. [9]Department of Biology, Ottawa Institute of Systems Biology, Carleton University, Ottawa, ON K1S 5B6, Canada. [10]Department of Microbiology, Faculty of Biological Sciences, Alzahra University, Tehran, Iran. [11]Department of Biotechnology, College of Science, University of Tehran, Tehran, Iran. [12]Department of Pathology and Molecular Medicine, Queens University, Kingston, ON K7L 3N6, Canada. [13]Department of Chemistry, The University of Chicago, Chicago, IL, USA. [14]Department of Immunology & Cell Biology, Université de Sherbrooke, Sherbrooke, QC J1E 4K8, Canada. [15]These authors contributed equally: Taha Azad, Reza Rezaei. ✉e-mail: jbell@ohri.ca

thought of as biological machines that to be optimally deployed, will require control strategies coordinating and regulating their activity within the patient[4,5]. Oncolytic viruses (OVs) have the potential to be replicating cancer gene therapy vectors that reproduce in and destroy tumor cells (oncolysis) and in addition have the capacity to dispense therapeutic payloads to remodel the tumor microenvironment[3] by impacting tumor vasculature[6] and reshaping extracellular matrix components[7]. However, this remodeling process, especially the generation of robust adaptive and innate immune responses, can sometimes be at odds with optimal virus spread within and between tumors. Incorporation of genetic circuits into oncolytic virus backbones that regulate the timing/magnitude of expression of virulence genes and therapeutic payloads could facilitate safer, more effective therapeutics. In this study, we used synthetic virology approaches to develop a new generation of OVs equipped with multiple inducible systems, which can be used as "safety switches" for providing user defined control of both virus replication and transgene expression. To facilitate clinical translation of our products we tested orally available FDA-approved small-molecule responsive transcription systems. We show through combinatorial application of discrete regulatory elements we can increase the safety and efficacy of an oncolytic vaccinia virus in xenograft and syngeneic mouse tumor models. These systems will pave the way for developing a generation of OVs in which the timing and level of both OV replication and their therapeutic payloads can be regulated.

## Results

### Generation of a rapamycin-inducible expression system in an oncolytic vaccinia virus vector

We designed a series of chemogenetic switches to enable temporal control of the expression of multiple therapeutic payloads. Vaccinia virus (VV) is a cytoplasmic virus and while is able to infect and replicate in mammalian cells, its transcriptional machinery and genomic structure is not optimized for the use of mammalian promoters. Vaccinia virus has its own set of regulatory sequences and transcription factors that are adapted to its specific needs, and the use of mammalian promoters is not compatible with these requirements (Supplementary Fig. 1a). Previous studies have demonstrated that the VV-encoded T7 polymerase can drive the expression of gene cassettes under the control of a T7 promoter element and so we incorporated a recently developed small-molecule activated, proximity-dependent split T7 RNA polymerase (RNAP)[8]. This system consists of T7 RNA polymerase divided into two domains (N term-1-29 and C term-30-181 of T7 RNAP), that assemble into a functional polymerase in a proximity-dependent fashion. The T7 components were fused to a rapamycin-induced dimerization system, consisting of FKBP-rapamycin binding domain (FRB) and FK506 binding protein (FKBP). Upon introduction of rapamycin, FRB is recruited to FKBP, which brings the C-terminus in close proximity to N-terminal T7 RNAP. T7 RNAP is then assembled and able to transcribe a gene regulated by a T7 promoter (Fig. 1a). We engineered VV to constitutively express mCherry and the rapamycin-controlled, split T7 RNA polymerase-based transcription system (ST7) in the Thymidine Kinase (TK) coding region. In the same TK locus, we also engineered a T7 promoter-driven firefly luciferase and GFP (GFPLuc) cassette (VV-ST7-iGFPLuc; Fig. 1a). In VV-ST7-iGFPLuc infected U2OS osteosarcoma cells, GFP and luciferase expression were induced dramatically in the presence of 10 nM rapamycin over a range of different virus concentrations (Fig. 1b, c). Infected U2OS cells showed a greater than tenfold increase in luciferase activity 24 h post infection relative to uninduced infected controls (Fig. 1c). Similar results were observed in HeLa cervical carcinoma cells, HT29 colorectal adenocarcinoma cell line, and A549 lung carcinoma cells (Fig. 1d). In order to confirm that rapamycin and the inducible split T7 polymerase system did not impact viral growth kinetics, we performed multistep virus growth curves. Rapamycin and ST7 expression did not have a significant effect

on VV-ST7-iGFPLuc growth in U2OS, HeLa, or A549 cells (Fig. 1e; Supplementary Fig. 1b, c).

Rapamycin (sirolimus) is approved for use as an immunosuppressant and treatment of lymphangioleiomyomatosis, a rare lung disease[9]. Its actions are mediated through inhibition of the mTOR protein kinase and associated with anti-proliferative and anti-cancer activity. Rapamycin suffers from poor solubility, stability and pharmacokinetic properties. However, rapamycin analogs, or "rapalogs", (e.g. everolimus, temsirolimus, and ridaforolimus) have been approved for the treatment of a wide spectrum of cancers[10,11]. We examined if these approved rapalogs were compatible with the VV-adapted ST7 expression system. In infected U2OS cells, we observed a robust increase in GFP and luciferase expression during treatment with rapalogs, in line with our observations with rapamycin (Fig. 1f–g). We also demonstrated a dose-dependent increase of luciferase activity in infected U2OS cells treated with rapalogs, suggesting the system is tunable (Fig. 1g). Similar to rapamycin, rapalogs did not significantly influence viral growth (Fig. 1h). To examine whether rapamycin and rapalogs can be used in vivo to control the virally encoded transgenic RNA polymerase system, we injected HT-29 xenograft bearing mice with VV-ST7-iGFPLuc intratumorally and treated mice with rapamycin or rapalogs via different routes of administration. Temsirolimus[12] is more water soluble and suited for intravenous injection, whereas everolimus[13] is available for oral administration. In this study, we infected tumors and applied a single dose of the rapalog at the time of infection using the clinically appropriate route for each rapalog. As shown in Fig. 1i-j, we observed robust stimulation of luciferase 24 h later. The animals received no further treatments with the rapalogs and then at 72 h post initial infection the tumors were once again imaged to determine the level of luciferase expression. At this time, the level of luciferase expression driven by the initial induction with the rapalog decreased significantly (Fig. 1i, j), This decrease in signal is not related to a clearance of virus as we determined the level of infectious particles at day 10 and it was comparable in all of the tumors sampled (Supplementary Fig. 1d). Furthermore, in animals that were infected but not treated with rapalog there was minimal luciferase signal which was robustly induced upon rapalog induction. Among the conditions tested, rapamycin (i.p.) and temsirolimus (i.v.) produced the highest induction of luciferase signal in the tumors. These results demonstrate that the rapamycin-inducible ST7 system can be used to temporally control viral transgene expression in vivo.

### Control of vaccinia virus promoter elements with doxycycline

We investigated the utility of repressor/operator systems to create virally encoded chemogenetic switches. Doxycycline (Dox) is an approved orally delivered tetracycline-class antibiotic with well-established pharmacokinetic and pharmacodynamic properties. The Dox-inducible expression system has previously been applied in the context of a VV genome to enable conditional expression of viral factors[14]. We carried out a more detailed analysis of this technology by applying it to a variety of vaccinia promoter elements. Vaccinia genes have stage-specific transcription factors and corresponding promoter elements which mediate temporal regulation of their expression. We screened a diverse set of VV promoters encompassing the range of temporal expression patterns exhibited by VV proteins. Using ribosome profiling in VV infected HeLa cells, we identified viral promoters associated with genes encoding transcripts that are efficiently translated, and thus highly expressed, at different times during the virus replication cycle (Fig. 2a). We generated a series of recombinant VV strains encoding tet operator controlled VV candidate promoters to conditionally control the expression of the GFPLuc cassette (Fig. 2b). In the same locus (TK), the strains encoded blue fluorescence protein (BFP) and the tetracycline repressor protein (TetR), which binds TetO in the absence of Dox to impair expression. We performed an arrayed screen of these viruses to identify which TetO-controlled VV promoter

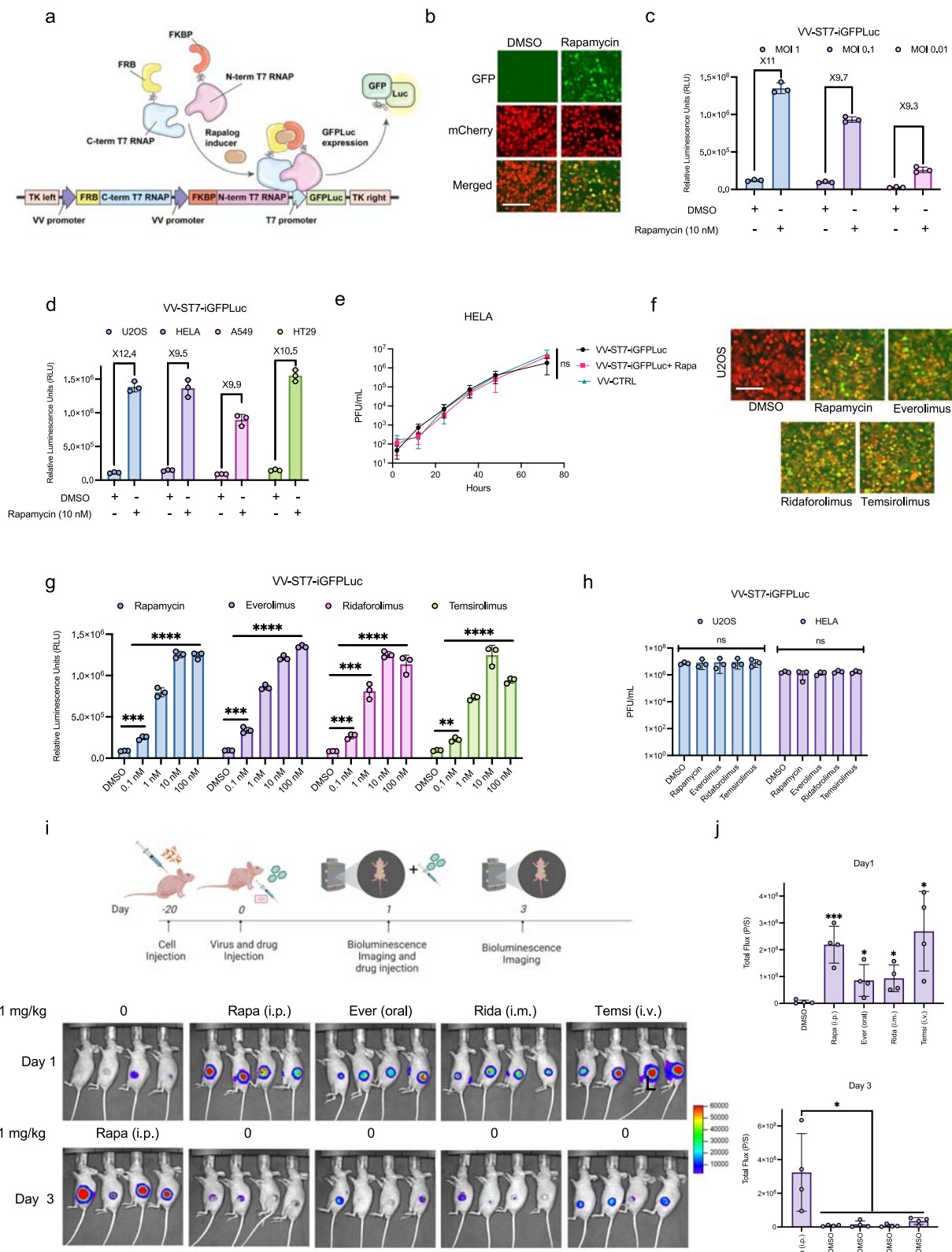

resulted in the highest level of induced GFP and luciferase expression (Fig. 2c, d). There was considerable variability in the ability to control vaccinia virus promoters using the Dox-inducible gene cassettes. We found that the p11 and PEL promoters were most amenable to this form of transcriptional control with robust induction in the presence of Dox and minimal background expression in its absence. A closer examination of the luciferase assay results (Fig. 2d) revealed that the use of

p11 promoter resulted in maximal reporter gene expression in the presence of Dox; therefore, we selected the TetO-controlled P11 promoter for further characterization studies. We first tested the kinetics of inducible expression with the newly developed VV strain encoding TetR and p11-TetO driven GFPLuc gene cassettes (VV-TetR-iGFPLuc). We observed induction of a GFP signal in infected U2OS cells within 3 h of Dox treatment (Supplementary Fig. 2a) and similar high inducibility

**Fig. 1 | Integration, functional characterization, and optimization of the split T7 inducible system in vaccinia virus. a** Schematic illustration of the split T7 inducible expression cassette inserted into Thymidine Kinase (TK) open reading frame. The FKBP linked N-terminal portion of the T7 RNAP and the FRB linked C-terminal portion of the T7 RNAP were expressed under continuous vaccinia virus promoters. A Fusion protein consisting of GFP, and firefly luciferase (GFPLuc) was incorporated under T7 promoter and induced by the addition of Rapalogs. mCherry fluorescent protein is expressed from the virus to detect virus infection in cells. **b, c** Representative fluorescent images and quantitation of luciferase signal (RLU) of the T7 inducible system from U2OS cells 24 h after infection with VV-ST7-iGFPLuc in the presence or absence of 10 nM rapamycin. mCherry indicates virus infection. **d** Comparison of the luciferase signal from various cell lines 24 h after infection at MOI 0.1 with VV-ST7-iGFPLuc in the presence or absence of 10 nM rapamycin. **e** Multistep growth curve of VV-ST7-iGFPLuc compared to the control vaccinia virus at different time points from Hela cells infected at MOI 0.01 in the presence or absence of 10 nM rapamycin. **f** Representative fluorescent images of the T7 inducible GFP system from U2OS cells 24 h after infection with VV-ST7-iGFPLuc at MOI 0.1 in the presence of different Rapalogs at 10 nM concentration. **g** Relative luminescence emitted after 24 h from U2OS cells infected with VV-ST7-iGFPLuc at MOI 0.1 in the presence of different concentrations of Rapalogs. **h** Comparison of vaccinia virus growth in the presence of Rapalogs at 10 nM in U2OS and HELA cells 24 h after infection at MOI 0.1. **i, j** IVIS imaging of HT-29 tumors in CD-1 nude mice. Tumors were injected with VV-ST7-iGFPLuc (1E7 PFU/tumor) when they reached ~150 mm³ in size. Different Rapalogs were administered at 1 mg/kg by their preferred route. After 24 h, luciferase signal was measured, and the control group received rapamycin intraperitoneally. Luciferase signal was measured in all groups again at 72 h. Scale bars = 40 μm in (**b, f**). Data indicate means ± SD of three (**c–e, g, h**) to four (**j**) biological replicates. ns $P > 0.05$, *$P < 0.05$ **$P < 0.003841$, ***$P < 0.000125$, ****$P < 0.001$ in unpaired two-samples $t$-test. Source data are provided as a Source Data file.

(>30-fold) in a panel of other cancer cell lines (A549, HeLa, HT-29, and SKOV3) (Fig. 2e and Supplementary Fig. 2b). We conducted a comparison of luciferase expression at various time points using the wild-type vaccinia virus promoter and the doxycycline-controlled promoter with and without induction and demonstrated that both promoter systems achieve a similar maximum level of reporter expression (Supplementary Fig. 2c). Neither expression of TetR, incorporation of the TetO-controlled expression system, or Dox influenced viral growth – illustrating compatibility of the Dox-controlled chemogenetic switch in VV-based vectors (Fig. 2f and Supplementary Fig. 2d, e). Under in vitro conditions, 10 ng/mL of Dox was sufficient to induce detectable levels of luciferase activity in VV-TetR-iGFPLuc infected U2OS cells, while 25 ng/mL produced the maximal signal (Supplementary Fig. 2f).

We then characterized VV-TetR-iGFPLuc in vivo using a HT-29 colon cancer xenograft model. We infected the mice intratumorally with VV-TetR-iGFPLuc and performed 1 mg/kg intraperitoneal treatments. In vivo imaging of bioluminescence, 24 h post infection, demonstrated robust luciferase activity in the tumors (Supplementary Fig. 2g). We repeated the experiment with one alteration—introduction of Dox via diet to simulate the potential use of oral Dox in combination with a VV vector bearing Dox-controlled transgene expression (Fig. 2g, h, Supplementary Fig. 2h). We were able to conditionally control transgene expression through feeding and withdrawal of a Dox-containing diet (Fig. 2h; Supplementary Fig. 2g). We noted that Dox had no influence on viral growth in vivo, in line with our in vitro results (Fig. 2g), and no negative effect on luciferase activity in vivo (Supplementary Fig. 2i, j). Collectively, these results establish the utility of a Dox-controlled gene cassette as a tool for temporal control of payload expression in VV vectors in vivo.

## A cumate-controlled chemogenetic switch for vaccinia virus

We sought to identify a repressor-based operator system to generate another VV-adapted inducible expression system to further enable independent temporal control of multiple payloads. The cumate-controlled operator system derived from the *Pseudomonas putida* p-cmt and p-cym operons[15,16], has been used in gene therapy vectors[17,18] and is similar to the tetracycline-controlled operator system. It consists of the cumate operator (CuO) sequence which is bound by a repressor (CymR) in the absence of cumate. We engineered VV to constitutively express CymR and BFP. In the same locus (TK), we incorporated the coding sequence of GFPLuc under the control of the cumate operon fused to a VV promoter (Fig. 3a). Similar to the development of the Dox-based chemogenetic switch, we screened VV promoters to identify which worked best in combination with the CuO to mediate cumate-inducible expression in infected U2OS cells (Fig. 3b, c). As with the Dox system, there was considerable variability in the control of vaccinia promoters by the cumate operon. While the early-late promoters H5R, LEO, and LEO160 resulted in maximal luciferase activity

and GFP expression in the presence of cumate, these promoters also resulted in the highest basal reporter signal in the absence of cumate. The use of the synthetic early-late promoter PEL in tandem with CuO resulted in the highest inducibility (450X) while maintaining minimal basal expression (Fig. 3c); therefore, we selected the PEL-Cu promoter/operator combination for further evaluation.

We examined the kinetics of inducible expression with our VV strain encoding CymR and PEL-CuO driven GFPLuc gene cassettes (VV-CymR-iGFPLuc). With cumate concentrations as low as 10 μg/mL, GFP signal is observed within 6 h of cumate treatment of VV-CymR-iGFPLuc infected cells (Supplementary Fig. 3a). Consistent with this, significant luciferase activity is also observed in infected cells with 10 μg/mL cumate (Supplementary Fig. 3d). We confirmed that cumate treatment does not influence viral growth in U2OS cells at concentrations up to 500 μg/mL (Supplementary Fig. 3b, c). Similarly, multistep viral growth studies confirmed that cumate or CymR expression does not impact VV replication in U2OS, HeLa, and A549 cells (Supplementary Fig. 3e–g). We next compared head-to-head the cumate-inducible expression system with the ST7 and Dox-inducible systems (Supplementary Fig. 3h–k). In infected U2OS cells, the Dox-inducible system produced the highest reporter signal in the presence of its inducer; however, the cumate-inducible system provided the greatest inducibility in the presence of its ligand while maintaining minimal basal expression in the absence of cumate (Supplementary Fig. 3h).

HT-29 xenografts were injected intratumorally with VV-CymR-iGFPLuc and cumate (1 mg/kg) was administered intraperitoneally. In vivo imaging of bioluminescence 24 h post infection revealed robust tumor-localized reporter signal in a cumate-dependent fashion (Supplementary Fig. 3l). In parallel, we demonstrated that cumate treatment in vivo did not influence VV replication (Supplementary Fig. 3l, lower panels). To examine the potential for more accessible administration of cumate, we repeated the experiment and delivered cumate (6000 mg/kg) to mice via their diet. We observed cumate-dependent activation of luciferase activity in the tumor, and, confirmed that cumate does not influence VV replication (Supplementary Fig. 3m, n). As this is the first report, to the best of our knowledge, of feeding mice cumate, we examined the effect of feeding cumate on mouse health and viral growth. At all concentrations tested (100–6000 mg/kg), cumate did not impact viral replication in vivo (Fig. 3d). Treatment for more than twenty-five days with a diet containing less than 2000 mg/kg cumate did not cause significant loss of weight in immunocompetent mice (Fig. 3e). For a more comprehensive toxicity analysis, we also examined the activity of the serum enzymes alkaline phosphatase, asparatate aminotransferase, and glutamate dehydrogenase. Twenty-five days of cumate diet, at all cumate concentrations tested (100-6000 mg/kg), did not impact on the levels of these enzymatic markers of liver toxicity (Fig. 3f and Supplementary Fig. 3o–r). The data suggests that the cumate-inducible system has an adequate safety profile for use in vivo.

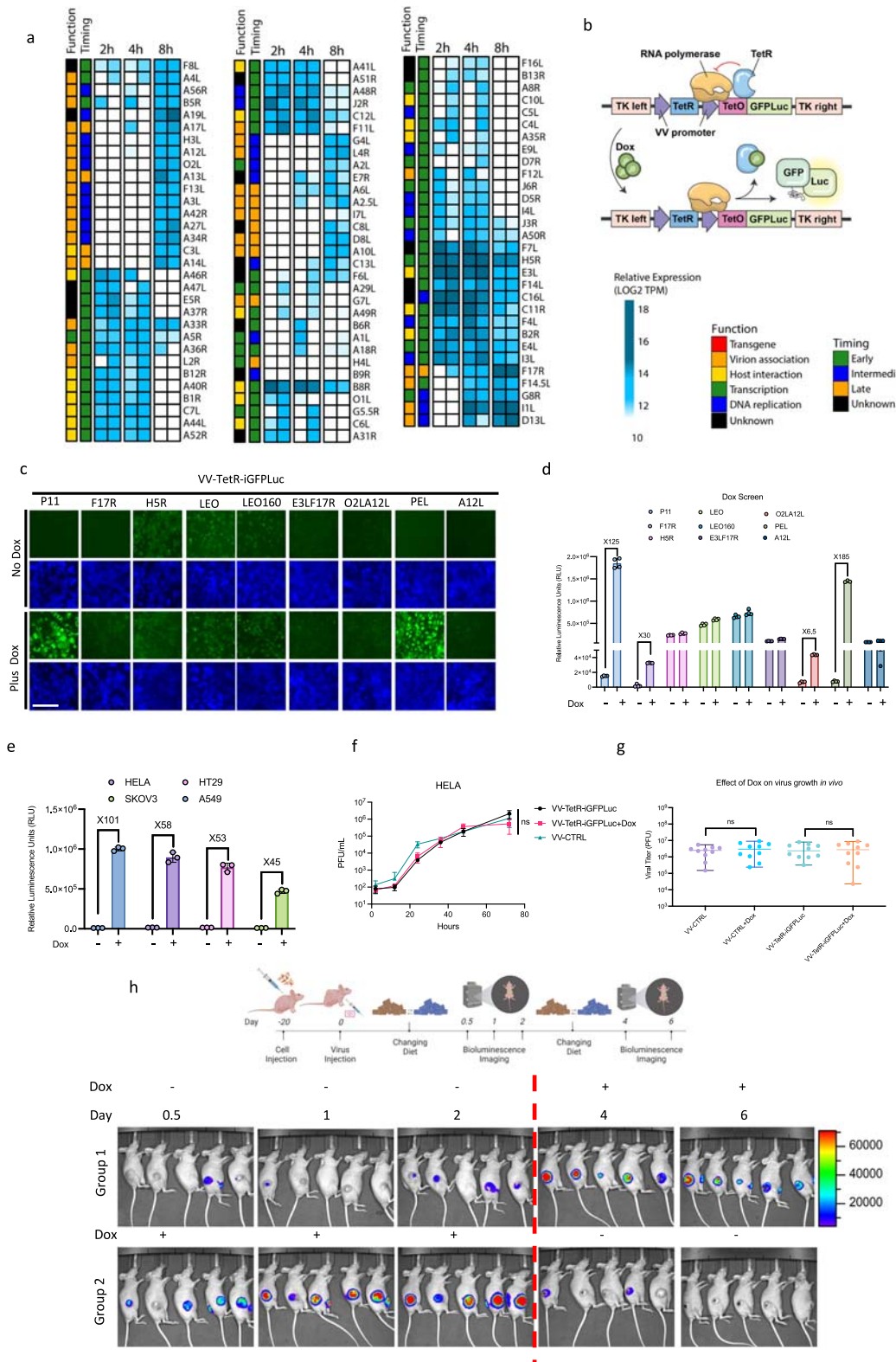

Lastly, we examined the sensitivity of the cumate-inducible system in vivo (Fig. 3g–j). We were able to rapidly induce luciferase expression in VV-CymR-iGFPLuc infected mice within 1 day of treatment. Upon removal of the cumate, the luciferase signals dissipated within 96 h. Overall, the data establish VV-CymR-iGFPLuc as a recombinant virus enabling tunable expression of a target transgene or viral factor.

**Combinatorial application of chemogenetic switches**

We noted that relative to the TetO or CuO controlled VV promoters, the VV-adapted T7 system resulted in higher basal expression of transgenes. We reasoned that combining the ST7 system with either operator would result in a more selective safety switch controlled by the use of two small molecules. We generated a virus encoding a TetO immediately downstream of a T7 promoter driving GFPLuc expression

**Fig. 2 | Optimization of the doxycycline-inducible system for controlling gene expression in vaccinia virus. a** HeLa cells were infected with vaccinia virus at MOI 10 and RNA was collected at 2, 4, and 8 h post infection. The heatmap, categorized by gene function and reported timing of gene expression, depicts results of the ribosome profiling after cDNA synthesis and sequencing. **b** Schematic illustration of the tetracycline inducible expression cassette inserted to TK locus. The TetR protein was expressed under a continuous vaccinia virus promoter and various vaccinia virus promoters preceding a TetO element, which binds TetR protein, were incorporated upstream of the GFPLuc fusion protein. The dissociation of the TetR from TetO upon doxycycline (Dox) administration leads to initiation of transcription and gene expression. Blue fluorescence protein (BFP) is continuously expressed from the virus to detect viral infection. **c, d** Representative fluorescent images and quantitation of luciferase signal (RLU) of U2OS cells 24 h after infection with viruses expressing the GFPLuc fusion protein (VV-TetR-iGFPLuc) at MOI 0.1 under the control of various native and synthetic vaccina promoters. Expression of GFPLuc was induced with 100 ng/ml Dox. BFP images indicate virus infection.

**e** Comparison of luciferase signal (RLU) from different cell lines 24 h after infection at MOI 0.1 with VV-TetR-iGFPLuc in the presence or absence of 100 ng/ml Dox. **f** Multistep growth curve of VV-TetR-iGFPLuc compared to the control vaccinia virus at different time points using Hela cells infected at MOI 0.01 in the presence or absence of 100 ng/ml Dox. **g** Quantitation of viral titers within HT-29 tumors from CD-1 nude mice seven days after Dox (625 mg/kg) was introduced into the diet of mice. Tumors were infected with control vaccinia virus (VV) or VV-TetR-iGFPLuc at 1E7 PFU/tumor prior to the introduction of Dox. **h** IVIS imaging of HT-29 tumors following injection with VV-TetR-iGFPLuc at 1E7 PFU/tumor when tumors reached ~150 mm³ in size. Mice were given Dox (625 mg/kg) in their diet 2 days post viral infection (Group 1) or immediately following viral infection (Group 2) of tumors. For Group 2, Dox was removed from the diet after 2 days. Luciferase signal was monitored at 12 h, 1, 2, 4 and 6 days post viral infection. Scale bars = 40 μm in (**c**). Data indicate means ± SD of three (**e, f**), four (**d**), or ten (**g**) biological replicates. ns $P > 0.05$, *$P < 0.05$ **$P < 0.003841$, ***$P < 0.000125$, ****$P < 0.001$ in unpaired two-samples $t$-test. Source data are provided as a Source Data file.

(Fig. 4a). The virus also encoded the TetR and ST7 (VV-ST7-TetR-iGFPLuc) to mediate Dox- and rapamycin-inducible expression of the reporter genes, respectively. In VV-ST7-TetR-iGFPLuc infected U2OS cells, we demonstrated that treatment with Dox alone increased luciferase activity 7-fold, whereas treatment with rapamycin alone induced luciferase activity by 19-fold (Fig. 4b). A combination of Dox and rapamycin treatment synergized to activate luciferase activity by 48-fold. Similar trends were observed using the GFP reporter system (Fig. 4c).

Analogous to VV-ST7-TetR-iGFPLuc, we developed a recombinant VV encoding ST7, CymR, and a CuO immediately downstream of a T7 promoter driving GFPLuc expression (VV-ST7-CymR-iGFPLuc; Fig. 4d). In VV-ST7-CymR-iGFPLuc infected U2OS cells, treatment with cumate alone, rapamycin alone, or both in combination induced luciferase activity 26-fold, 54-fold, and 97-fold respectively (Fig. 4e, f). Importantly, incorporation of either the TetO or CuO operators into the ST7 system significantly reduced basal "leaky" expression associated with the ST7 system as illustrated by the complete loss of basal luciferase activity in the absence of small-molecule inducers.

During our generation of a VV-adapted cumate-inducible expression system, we noted that while H5R, LEO, and LEO160 promoters produce the strongest luciferase activity in the presence of cumate (Fig. 3c), but also produced the highest basal expression in the absence of cumate. We hypothesized that combining CuO with TetO would produce a more tightly regulated expression system (Cu-Tet double inducible expression system). We engineered seven different recombinant VV expressing CymR, TetR, as well as a CuO and TetO placed immediately downstream of unique VV promoters (p11, H5R, LEO, LEO160, O2LA12L, E3LF17R and P11-NS) driving GFPLuc expression (Fig. 4g). Of the VV promoters screened, LEO in concert with a TetO and CuO produced the highest inducible luciferase activity in the presence of Dox and cumate, while maintaining low basal activity in the absence of small-molecule inducers (Fig. 4h) in infected U2OS cells. Treatment with Dox alone or cumate alone induced intermediate levels of luciferase activity. Analyses of GFP fluorescence revealed similar trends (Fig. 4i). We utilized VV expressing CymR, TetR, and CuO-Tet-O-LEO driven GFPLuc (VV-CymR-TetR-iGFPLuc) for additional in vitro characterization studies. To confirm that this level of tunability was not specific to U2OS cells, we examined the inducibility of luciferase activity in VV-CymR-TetR-iGFPLuc infected HEK293, Vero, HT-29, 786-0, and HeLa cells (Fig. 4j). In all tested cell lines, Dox induced 3- to 7-fold increase in reporter signal, whereas cumate induced a 11- to 23-fold activation of luciferase activity. Combined treatment with Dox and cumate produced maximal increases in luciferase activity ranging from 23- to 45-fold across the five tested cell lines. As a proof-of-concept, we also demonstrated that a 3rd generation Dox-inducible expression system (Tet-ON using reverse tetracycline-controlled transactivator 3, rTA3) can be combined with the cumate-inducible

expression system (CymR/CuO) to generate a highly tunable expression system in an oncolytic herpes virus platform (HSV: Fig. 4k–m)—illustrating that the chemogenetic switches developed for VV have the potential to be adapted for other viruses.

The development of the VV-CymR-TetR, VV-ST7-TetR and VV-ST7-CymR expression systems enable the introduction of additional tunable functionality into poxviruses. Further, these inducible expression systems enable multiplexed control of multiple virally encoded transgenes through the addition of various combinations of rapamycin, cumate, and doxycycline.

## Chemically regulated control of oncolytic virus replication and spread

Oncolytic viruses with deletions in critical virulence genes have a very strong safety record but this often comes at the price of compromised therapeutic potency and challenges to manufacturing. An alternative strategy for selective replication of an OV is to control expression of key replication gene products. Recent studies have described that deletion of the D13L gene in VV produces a severely attenuated vector incapable of producing infectious virus in healthy volunteers[19,20]. Thus, we replaced the endogenous vaccinia virus D13L promoter with a TetO-controlled p11 promoter. (VV-TetR-iD13; Fig. 5a). As expected, in VV-TetR-iD13 infected U2OS cells, the addition of Dox increased plaque size, as observe by GFP fluorescence (Fig. 5b) and Dox dependent cell killing (Supplementary Fig. 4a). These results were observed in a panel of infected human cancer cell lines of a variety of histologies (A549, HeLa, HT-29, SKOV3: Fig. 5c, d and Supplementary Fig. 4b, c). While vaccinia replication as measured by a multistep growth curve analysis was severely compromised in the absence of Dox, the growth rate of VV-TetR-iD13L was indistinguishable from wild-type vaccinia virus in the presence of the inducer (Fig. 5e, f). Furthermore, our observations indicate that this virus replicates 100–1000 times less efficiently in human primary cells when compared to cancer cells (Supplementary Fig. 4d).

We next examined the temporal control of VV-TetR-iD13 in HT-29 xenograft bearing mice in vivo. Bioluminescence imaging of HT-29 xenografts revealed that in the presence of Dox-containing diet, robust tumor selective viral replication was observed within 12 h of viral infection (Fig. 5g–h). However, after changing back to normal diets viral replication was extinguished. Similarly, it was possible to delay the introduction of a Dox-containing diet to 2 days post infection, and robust tumor-localized luciferase activity would be observed 4–6 days post infection. Collectively, the data demonstrates that the chemogenetic switches (e.g., TetR/TetO) can be applied as a safety switch to generate VV vectors with chemically regulated replication kinetics in vivo.

While our primary interest is devising strategies to manipulate oncolytic virus replication and transgene expression, it seems

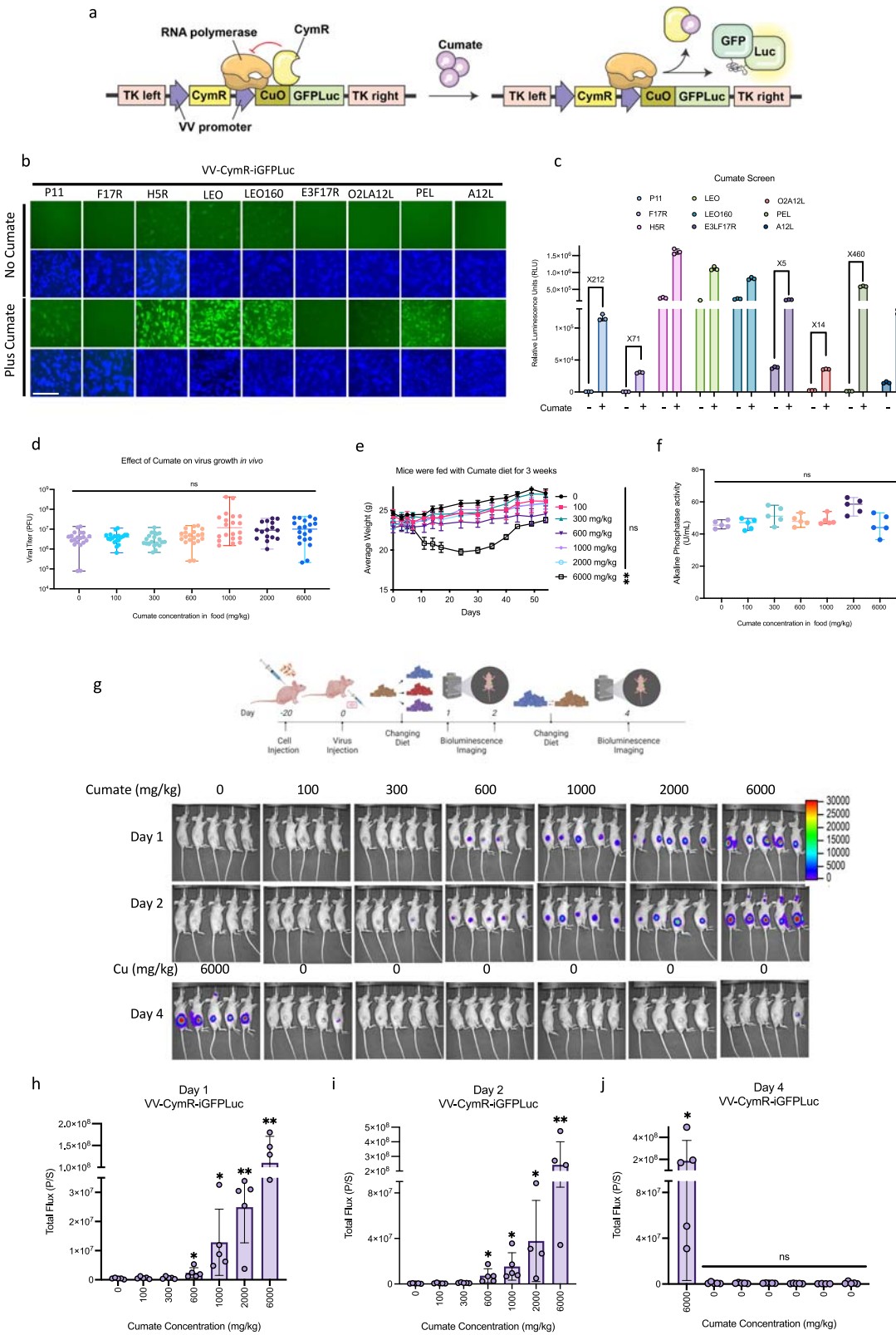

reasonable to expect that conditional virus gene expression could be also used to create effective vaccine vectors for either infectious diseases or certain cancer indications. The use of replication competent VV-vectored vaccines in the clinic to-date has been limited due to potential adverse events, including progressive vaccinia and post-vaccination encephalitis. We hypothesized that the use of the conditionally replicating VV-TetR-iD13 as a backbone vector for

immunogen delivery could help reduce safety concerns. To examine the safety of VV-TetR-iD13, we compared the effects of VV-TetR-iD13 and VV-CTRL infection in SCID mice after intravenous injection. VV-TetR-iD13 infection was followed by two days of Dox treatment to allow for initial replication to enhance any antigen expression in the context of a vaccine vector. Five days post infection, we noted a significant number of pox lesions in VV-CTRL infected mice; however,

**Fig. 3 | Cumate-inducible system controls viral gene expression in vitro and in vivo. a** Schematic illustration of the cumate-inducible expression cassette inserted into TK locus. The CymR protein was expressed under a continuous vaccinia virus promoter and a GFPLuc fusion protein incorporated under various vaccinia virus promoters preceding a CuO element which binds CymR protein. The dissociation of the CymR from CuO upon cumate administration leads to initiation of transcription and gene expression. Blue fluorescence protein is expressed from the virus to monitor its presence in infected cells. **b, c** Representative fluorescent images and quantitation of luciferase signal (RLU) from U2OS cells 24 h after infection at MOI 0.1 with viruses expressing the GFPLuc fusion protein (VV-CymR-iGFPLuc) under the control of various native and synthetic vaccinia promoters. Expression of GFPLuc was induced with 100 μg/ml cumate. **d** Quantitation of viral titers within HT-29 tumors from CD-1 nude mice seven days following infection with VV-CymR-iGFPLuc (1E7 PFU/tumor) and treatment with varying amounts of cumate.

**e, f** C57BL/6 mice were fed with varying amounts of cumate in their diet for 25 days and weighed at regular intervals for up to 55 days to measure cumate toxicity. Serum was collected at days 25 and 55 to measure different toxicity indicators. **g–j** HT-29 tumors from CD-1 nude mice were infected with VV-CymR-iGFPLuc (1E7 PFU/ml) and mice fed cumate diets according to the schedule and amounts shown. Bioluminescence images were taken, and luciferase activity quantified 1 and 2 days following initial virus treatment. After day 2, cumate diets were switched to a normal rodent diet and the control group received a diet containing 6000 mg/kg of cumate. Additional images were acquired following the diet switch, at day 4 post virus infection. Bar graphs show the total flux signal measured in the tumor area at the indicated days. Scale bars = 40 μm in (**b**). Data indicate means ± SD of three (**c**), twenty (**d**), and five (**e**–**j**) five biological replicates. ns $P > 0.05$, *$P < 0.05$ **$P < 0.003841$, ***$P < 0.000125$, ****$P < 0.001$ in unpaired two-samples $t$-test. Source data are provided as a Source Data file.

VV-TetR-iD13 infected mice had drastically lower number of lesions (Fig. 6a, d). This result was consistent with bioluminescent imaging of virus localization (Fig. 6b, e). Whereas in the VV-TetR-iD13 infected mice, there was no detected signal, the VV-CTRL infected mice had bioluminescent signals localized to tails, paws, and snouts. Furthermore, VV-CTRL infected mice demonstrated increasing weight loss from 4 to 8 days post infection (Fig. 6c). Virus biodistribution in known mouse VV reservoirs were analyzed via plaque assay and there was more a than tenfold decrease in viral titers in lungs, livers, and spleen. Collectively, the data demonstrate that conditionally replicating VV-TetR-iD13 represents a safer vector for antigen delivery.

We also evaluated the immunogenicity of VV-TetR-iD13 delivering a clinically relevant immunogen. The SARS-CoV-2 spike glycoprotein (VV-S-TetR-iD13) or a prefusion-stabilized spike variant (VV-S-HexaPro-TetR-iD13) was encoded into VV-TetR-iD13 to generate SARS-CoV-2 vaccine candidates. We confirmed that these viruses maintained Dox-inducible replication (Supplementary Fig. 5a, b) and expressed their respective antigens (Supplementary Fig. 5c). Subsequently, we analyzed the humoral responses induced in mice infected with these vaccine candidates in the presence of 48 h Dox treatment. A single dose of VV-S-TetR-iD13 or VV-S-HexaPro-TetR-iD13 was sufficient to induce robust Spike-targeted IgG response (Fig. 6f). However, an examination of neutralizing antibody titer levels via SARS-CoV-2 S pseudovirus neutralization assay suggested that HexaPro more consistently elicited a stronger response (Fig. 6g). In the absence of Dox treatment, there was a detectable, but significantly impaired antibody response elicited by VV-S-HexaPro-TetR-D13 (Fig. 6h, i). Further, we confirmed that VV-S-HexaPro-TetR-D13 induced higher antibody responses relative to a non-replicating vaccinia virus vector (MVA) delivering the same antigen (Fig. 6j, k). Taken together, these results demonstrate that chemogenetic switches can be applied to increase the safety of replicating vaccine VV vectors while maintaining their ability to induce strong immunogen specific antibody responses.

## Chemogenetic control of virus directed membrane fusion

There is accumulating evidence that encoding membrane fusion proteins into an oncolytic virus backbone can improve therapeutic activity by enabling neutralizing antibody-resistant viral spread, immunogenic cell death, and minimizing the release of virus into healthy tissue or systemic circulation[21]. However, expression of heterologous fusogenic proteins can also be counter-productive to OV replication, impacting virus manufacturing and in some circumstances causing off-target toxicity[22–25]. We hypothesized that integration of a chemogenetic switch to mediate temporal control of fusogenic protein expression would overcome some of these limitations. To identify an optimal fusion protein, we performed an overexpression screen in HEK293 cells of arenavirus and reovirus fusion proteins. To this end, HEK293 cells expressing GFP were transfected with a mammalian expression plasmid encoding a range of fusion proteins and then co-cultured with HEK293 cells stably expressing mCherry. Twenty-four hours post

transfection cells were analyzed for co-localization of mCherry and GFP (indicative of fusion between heterotypic cells). (Fig. 7a; Supplementary Fig. 6a). The screen revealed that two fusion proteins produced significant amounts of syncytia—the Tamiami virus glycoprotein (TAMV-GP) and reovirus p14 in a variety of cell lines (U2OS, Vero, A549 and HeLa, Supplementary Fig. 6b). We performed an Alamar Blue assay to analyze the effects of the GPs on cell viability (Fig. 7b). In a panel of cell lines (HEK293, U2OS, Vero, A549, and HeLa), overexpression of TAMV-GP or p14 resulted in a greater than 70% reduction of cell viability, whereas overexpression of p15, a reovirus fusion protein produced low levels of syncytia and resulted in minimal cell death (Fig. 7b). Previous work has demonstrated that expression of p14-induced cell fusion results in increased caspase 3 activation[25]. Consistent with this, we observed increased caspase activity in cells overexpressing p14 or TAMV-GP (Supplementary Fig. 6c). As both TAMV-GP and p14 showed robust induction of fusion and cell death, we generated recombinant VV strains encoding each fusion protein under the control of our Dox-inducible expression system (VV-TetR-iP14 and VV-TetR-iTAMV-GP; Supplementary Fig. 6d). Vaccinia virus has a complex replication cycle with multiple viral isoforms that contribute to its ability to disperse throughout an infected individual. To examine the contribution of membrane fusion to local virus spread within the tumor microenvironment, we analyzed virus-induced syncytia formation in the presence of ST-246, an established inhibitor of extracellular isoforms of orthopoxvirus[26] (Fig. 7c). While both viruses induced robust cell fusion in all tested cell lines in the presence of Dox, the addition of ST-246 significantly impaired the ability of VV-TetR-iP14 to induce syncytium. On the other hand, VV-TetR-iTAMV-GP's ability to induce syncytia was not impacted by ST-246 treatment – suggesting VV-TetR-iTAMV-GP, in the presence of Dox treatment, has an enhanced ability to spread via cell fusion relative to VV-TetR-iP14. We analyzed the impact of TAMV-GP expression on the growth of the recombinant virus. Multistep viral growth curve analysis in A549 and HeLa cells showed that VV-TetR-iTAMV-GP infection with simultaneous Dox treatment results in decreased viral growth (Fig. 7d, e); however, delaying expression of TAMV-GP by treating with dox 48 h post infection resulted in similar viral growth kinetics to untreated VV-TetR-iTAMV-GP infected cells. This suggests that while constitutive expression of the fusion protein impairs viral growth, the use of a chemogenetic switch will enable temporal control of TAMV-GP expression allowing productive virus infection of the tumor before initiating membrane fusion and its downstream therapeutic sequelae. To test this hypothesis, we evaluated the anti-tumor effects of VV-TetR-iTAMV-GP in vivo using the A549 xenograft model infected in the presence or absence of Dox (Fig. 7f–i). Without a Dox-containing diet, VV-TetR-iTAMV-GP had decreased anti-tumor activity which was significantly enhanced in mice fed a Dox-containing diet 48 h post infection. Tumor volumes and weight were significantly reduced in mice co-treated with VV-TetR-iTAMV-GP and Dox relative to the virus alone. Immunohistochemistry analysis of tumors revealed Dox

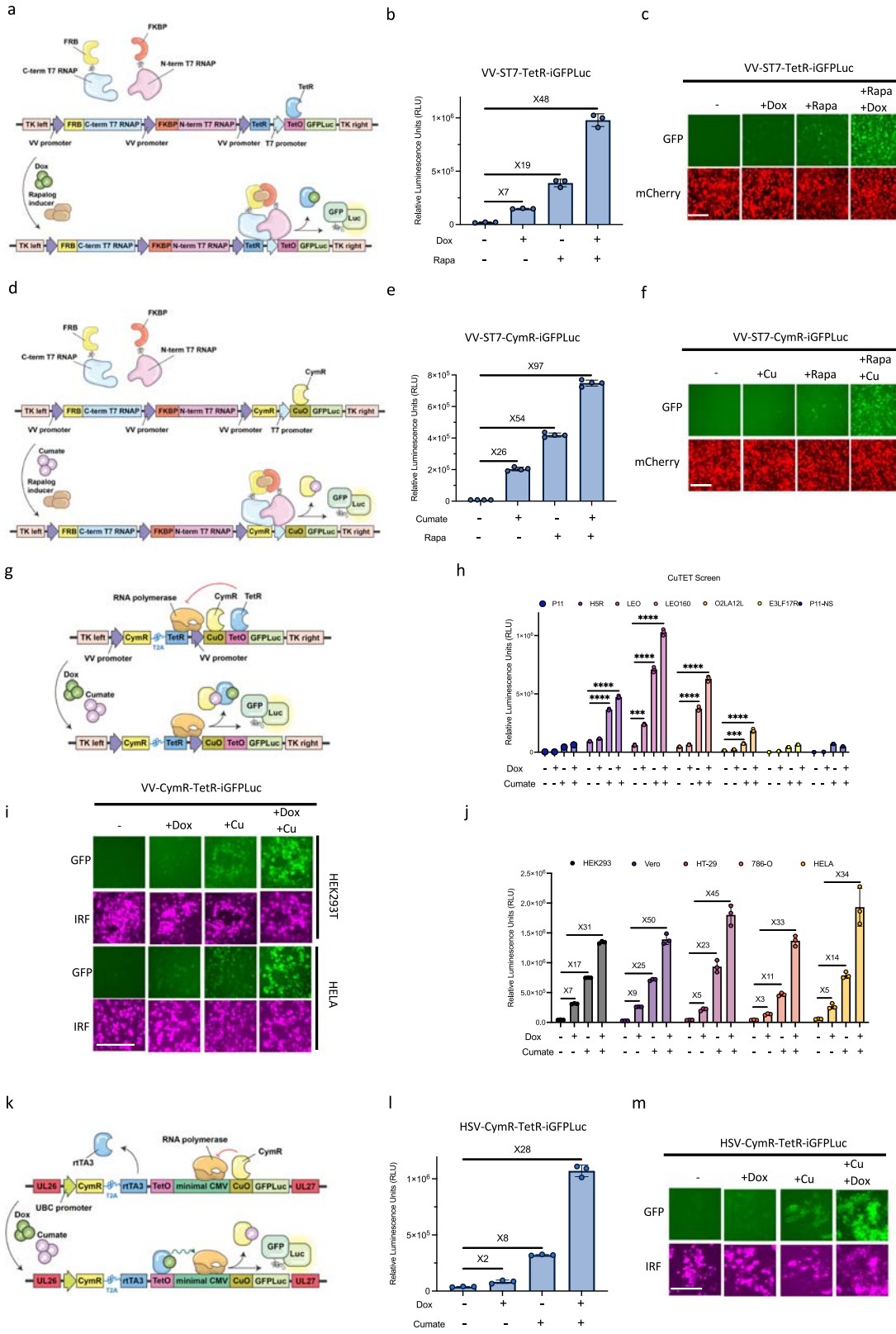

treatment-specific induction of TAMV-GP and caspase activation in virus infected cells, consistent with our in vitro results (Fig. 7i, Supplementary Fig. 6e). We repeated the study utilizing intravenous delivery, and our results demonstrated that systemic delivery of VV-TetR-iTAMV-GP in the presence of a Dox-containing diet was superior to virus alone (Supplementary Fig. 6f–h). Collectively, our results demonstrate that the application of chemogenetic switches in VV

enables the chemically regulated delivery of virus-inhibitory payloads strategically delaying transgene expression while optimizing therapeutic effect.

## Orchestrating transgene expression for safety and efficacy
Several studies have attempted to deliver cytokines using viral vectors[27,28], however, for a subset of potent, pro-inflammatory

**Fig. 4 | Double inducible chemogenetic switches overcome major challenges with single molecule systems. a** Schematic illustration of the ST7-Tet double inducible expression cassette inserted into TK locus. The TetR and split T7 fragment proteins were expressed under continuous vaccinia virus promoters and a GFPLuc fusion protein was incorporated under T7 promoter preceding a TetO element which binds TetR protein. The binding of split T7 fragments in the presence of Rapalogs and dissociation of the TetR from TetO upon Dox administration leads to initiation of transcription and gene expression. **b, c** Representative fluorescent images and quantitation of luciferase activity from U2OS cells 24 h following infection with VV-ST7-TetR-iGFPLuc in the presence or absence of Dox and rapamycin. mCherry signal indicates virally infected cells. **d** Schematic illustration of the ST7-Cumate double inducible expression cassette inserted into the same locus as described in A. The CymR and split T7 fragment proteins were expressed under continuous vaccinia virus promoters and GFPLuc was incorporated under T7 promoter preceding a CuO element which binds CymR protein. The binding of split T7 fragments in the presence of Rapalogs and dissociation of the CymR from CuO upon cumate administration leads to initiation of transcription and gene expression. **e, f** Representative fluorescent images and quantitation of luciferase activity from U2OS cells 24 h following infection with VV-ST7-CymR-iGFPLuc in the presence or absence of cumate and rapamycin. **g** Schematic illustration of the Cu-Tet double inducible expression cassette inserted into the same locus as described in A. The CymR and TetR proteins were expressed under continuous vaccinia virus promoters and GFPLuc was incorporated under different vaccinia virus promoters preceding a CuO element which binds CymR protein, and a TetO element which

binds TetR protein. Dissociation of the CymR from CuO and TetR from TetO upon cumate and Dox administration leads to initiation of transcription and gene expression. **h** Luciferase signal (in RLU) was quantified 24 h after infection of U2OS cells with viruses expressing GFPLuc under the control of CuTet and various native and synthetic vaccinia promoters in the presence or absence of 100 μg/ml cumate and 100 ng/ml Dox. **i, j** Representative fluorescent images and quantitation of luciferase activity from HEK293T, Hela, Vero, HT-29, 786-O cells infected with VV-CymR-TetR-iGFPLuc in the presence or absence of cumate and Dox after 24 h. IRF signal indicates virally infected cells. **k** Schematic illustration of the CuTet double inducible expression cassette inserted in the intergenic location between UL26 and UL27 open reading frames of the HSV-1 genome. The CymR-T2A-rtTA3 fusion protein was expressed under the UBC promoter. For expression of the GFPLuc fusion protein, TetO followed by a minimal CMV promoter and CuO was used. In the presence of Dox, rtTA3 binds to the TetO and enhances CMV promoter activity which will be at maximum activity if CymR is also dissociated from CuO upon cumate administration. IRF is continuously expressed from the virus, to monitor viral infection. **l, m** Representative fluorescent images and quantitation of luciferase activity from U2OS cells infected with VV-CymR-TetR-iGFPLuc in the presence or absence of cumate and Dox after 24 h. IRF signal indicates virally infected cells. Scale bars = 40 μm in (**c, f, i, m**). Data indicate means ± SD of three (**b, h, j, l**) or four (**e**) biological replicates. ns $P > 0.05$, $*P < 0.05$ $**P < 0.003841$, $***P < 0.000125$, $****P < 0.001$ in unpaired two-samples $t$-test. Source data are provided as a Source Data file.

cytokines, the safety and efficacy of cytokine therapy is heavily influenced by the timing and intensity of dosage. Our chemogenetic switches provide an opportunity to adjust both *timing* and *magnitude* of payload expression. For these experiments we used the cumate-based operator as we had established the dynamic range of the VV-adapted cumate-regulated expression system in vivo (Fig. 3c–j; Supplementary Fig. 3h). Numerous lines of evidence indicate that IL-12, IL-2 and IL-18 give complementary immunoregulatory signals[29–32]. It has been demonstrated that these three cytokines exert synergistic anti-tumor action in preclinical tumor models by activating the innate immune response, increasing the cytotoxicity of natural killer cells, natural killer T cells and granulocytes, and potently shutting down the tumor vasculature and reversing the immune suppression mediated by dysfunctional myeloid cells in the tumor[29,33–35]. We generated several different PEL-CuO driven gene cassettes for combinatorial expression of the three cytokines, including a multi-cistronic construct with internal ribosome entry sites (VV-CymR-iIL2IRES-2-DIRES-18; Supplementary Fig. 7a), 2A self-cleaving peptide-based multi-gene expression system (VV-CymR-iIL12-P2A-2-T2A-18; Supplementary Fig. 7b), as well as independent PEL-CuO promoters driving each cytokine (Fig. 8a; VV-CymR-iIL12-2-18). Of the three tested systems, ELISA analyses revealed that three separate promoter systems resulted in the most robust, cumate-inducible expression of all three cytokines (Fig. 8b, Supplementary Fig. 7c). All three cytokines are known to activate IFN-γ and TNF-α expression in immune cells and so we cultured mouse splenocytes with the virus-filtered supernatants of VV-CymR-iIL12-2-18 infected U2OS cells and examined the levels of both cytokines in splenocyte supernatants. In the presence of cumate, VV-CymR-iIL12-2-18 infected cells' supernatant produced robust increases in IFN-γ and TNF-α production (Fig. 8c, d). The functional role of each cytokine was examined through the use of neutralizing antibodies against IL-2, IL-12 or IL-18. Blocking of IL-2 or IL-12 in the supernatants impaired activation of IFN-γ and TNF-α production while blocking of IL-18 only impaired IFN-γ production from splenocytes. These in vitro results demonstrated that VV-CymR-iIL12-2-18 encodes a cumate-regulated multi-expression system for IL-2, IL-12, and IL-18.

We examined whether the chemical regulation of cytokine expression by VV-CymR-iIL12-2-18 circumvents the toxicity associated with systemic administration of these cytokines. For our toxicity studies, we utilized CD-1 nude mice to mimic an immunocompromised cancer patient, who may be more susceptible to cytokine-induced

immunopathology due to persistence of the VV vector after administration. Whereas infection with VV-CTRL or VV-CymR-iIL12-2-18 in the absence of cumate resulted in no mouse mortality after 5 days, infection with VV constitutively expressing the three cytokines (VV-IL12-2-18) resulted 100% mortality (Fig. 8e). This is consistent with constitutive high levels of expression of all three cytokines being toxic. 80% mortality was observed in the mice infected with VV-CymR-iIL12-2-18 and fed a 6000 mg/kg cumate diet to induce comparable levels of cytokine production in mouse sera (Fig. 8f). Similar to VV-IL12-2-18, the 6000 mg/kg cumate combined with VV-CymR-iIL12-2-18 infection induced lung edema (Fig. 8g), and liver toxicity as evidenced by increases in serum alkaline phosphatase activity levels (Fig. 8h). In order to determine whether cumate dosage could be used to limit toxicity of VV-CymR-iIL12-2-18, we fed infected mice with a range of cumate doses (1000 mg/kg, 2000 mg/kg and 6000 mg/kg). At 1000 mg/kg cumate, we observed no mortality after 5 days in VV-CymR-iIL12-2-18 infected CD-1 nude mice (Fig. 8i). The 1000 mg/kg cumate diet induced a lower level of cytokine production in infected mice relative to the other tested doses and caused no significant differences in lung edema (Fig. 8k), or elevated liver markers (serum alkaline phosphatase or asparatate aminotransferase activity) (Fig. 8l, m). These results demonstrate that 1000 mg/kg is safe dosage of cumate in VV-CymR-iIL12-2-18 infected mice. Finally, we examined the anti-tumor effects of the combination of VV-CymR-iIL12-2-18 and 1000 mg/kg cumate in immunocompetent mice with peritoneal murine cancer (MC38) (Fig. 8n). As expected, mice treated with VV constitutively expressing the three cytokines while providing some limited benefit over VV lacking the transgenes, ultimately succumbed likely due to toxicity of the three cytokines. In contrast, with controlled cytokine expression using the VV-CymR-iIL12-2-18 virus and 1000 mg/ kg cumate we observed increased mouse survival from the tumor challenge to 40%. These results demonstrate that the chemogenetic switch enables dosage control to mediate delivery of potentially toxic payloads at safe and efficacious levels.

Finally, we demonstrated that both virus replication and payload delivery can be placed under separate controls to increase safety of the VV vector (VV-CymR-iIL12-2-18/TetR-iD13). This vector replicates conditionally in the presence of Dox (Fig. 8o) and expresses the three cytokines in the presence of cumate (Fig. 8p). It should be noted that both cumate and dox treatment were required for VV-CymR-iIL12-2-18/TetR-iD13 infected cells to produce the same levels of cytokine as

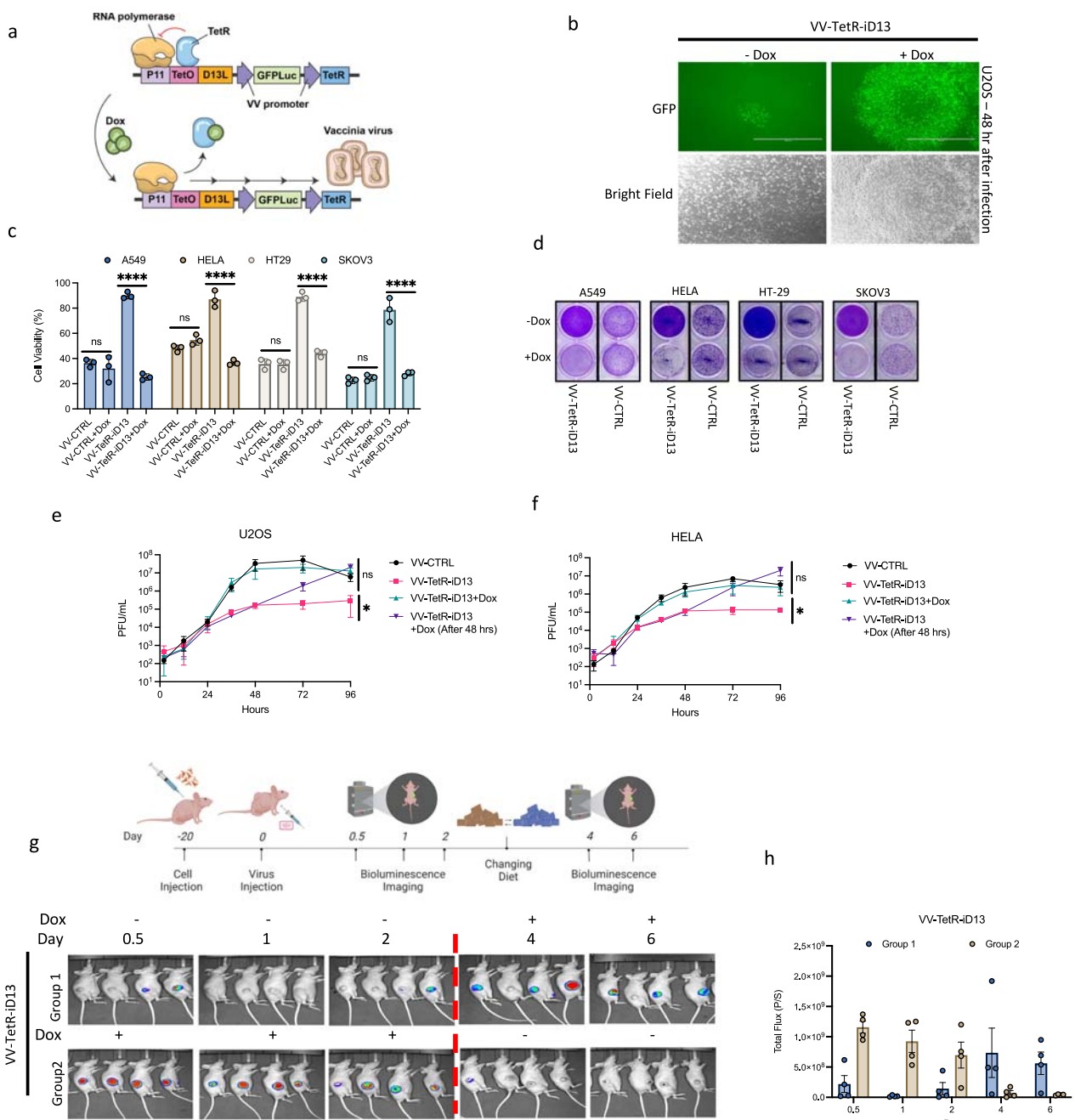

**Fig. 5 | Application of the tetracycline inducible system as a safety switch to generate a conditionally replicating vaccinia virus. a** Schematic illustration of the tetracycline inducible system controlling the expression of the D13L vaccinia virus gene which leads to the conditional growth of the vaccinia virus in the presence of Dox. **b** Representative images of U2OS cells 48 h after infection with VV-TetR-iD13 (MOI 0.01) in the presence or absence of 100 ng/ml Dox. **c, d** A549, Hela, HT-29, and SKOV3 cells were infected with control vaccinia virus or VV-TetR-iD13 at MOI 0.01 in the absence or presence of Dox at 100 ng/ml. After 48 h, cells were stained with crystal violet, and cell viability assessed by measuring absorbance at 570 nm using resazurin. **e, f** Multistep growth curves of VV-TetR-iD13 compared to control vaccinia virus at different time points from U2OS and Hela cells when

infected at MOI 0.01 in the presence or absence of 100 ng/ml Dox. **g, h** IVIS imaging of HT-29 tumors in CD-1 nude mice following injection with VV-TetR-iD13 (1E7 PFU/tumor) when tumors reached 150 mm³. Mice were given a Dox diet either 2 days post viral infection (Group 1, blue bars) or immediately following viral infection (Group 2, brown bars). For Group 2, Dox was removed from the diet after 2 days. Luciferase signal was measured at 12 h, 1, 2, 4 and 6 days after virus and drug administration. Bar graphs show the average total luciferase signal emitted in the tumor area. Scale bars = 200 μm in (**b**). Data indicate means ± SD of three (**c, e, f**) or four (**h**) biological replicates. ns $P > 0.05$, *$P < 0.05$ **$P < 0.003841$, ***$P < 0.000125$, ****$P < 0.001$ in unpaired two-samples t-test. Source data are provided as a Source Data file.

VV-CymR-iIL12-2-18 in the presence of cumate. This demonstrates that Dox-conditional replication adds another level control on payload delivery in addition to dox-inducible replication. Lastly, we examined the anti-tumor effects of VV-CymR-iIL12-2-18/TetR-iD13 in the murine model with peritoneal MC38. In line with the results obtained with 1000 mg/kg cumate and VV-CymR-iIL12-2-8 (Fig. 8n),

VV-CymR-iIL12-2-18/TetR-iD13 infection with induced replication (Dox treatment) and payload delivery (cumate treatment) resulted in 40% mouse survival (Fig. 8q). Taken together, our results demonstrate the application of two chemogenetic switches in VV enables the independent chemical regulation of both replication and payload expression.

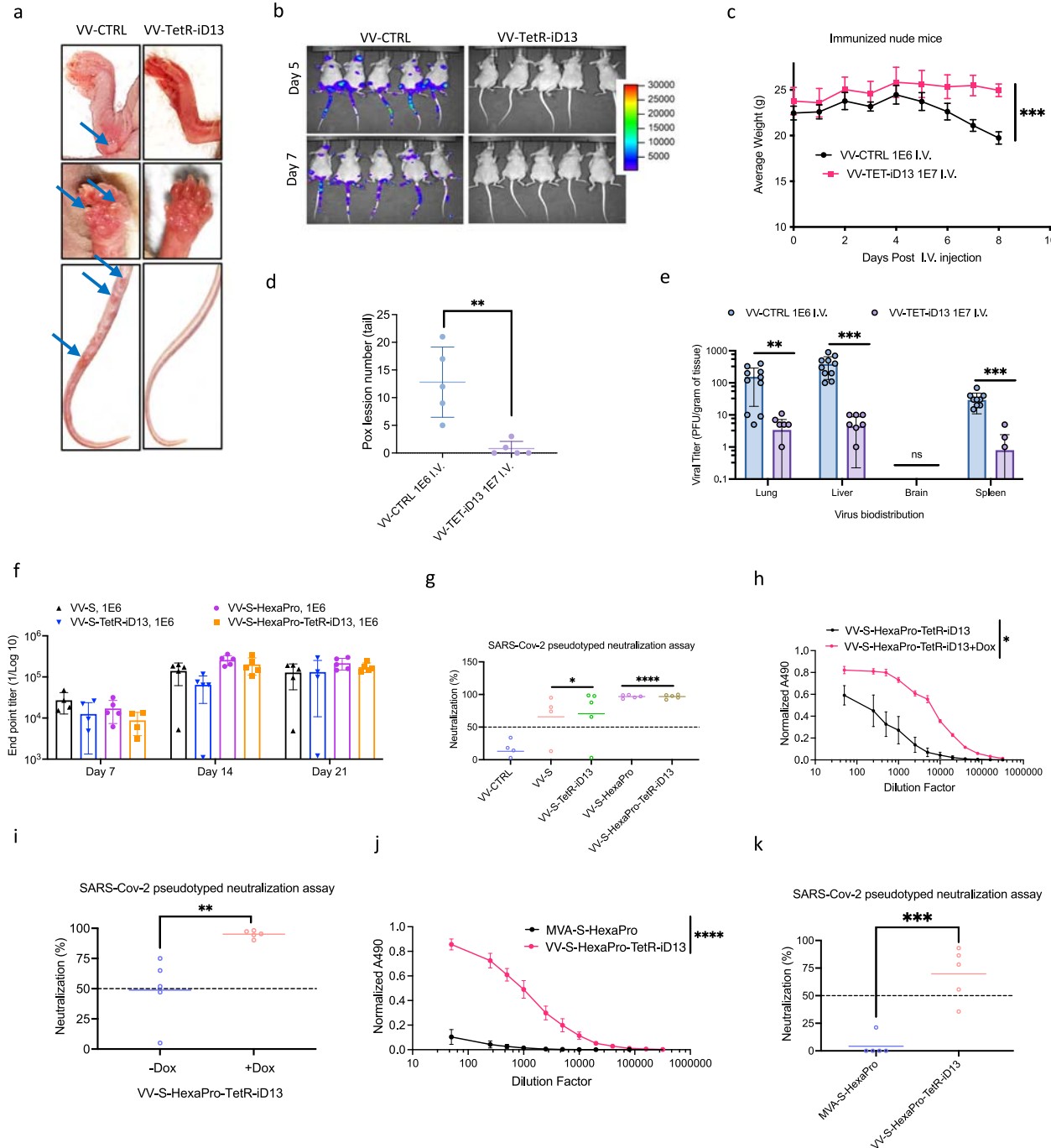

**Fig. 6 | Conditionally replicating vaccinia virus as a safe alternative for vaccine development. a–e** Evaluation of the safety of the conditionally replicating virus. Mice were injected intravenously with either control vaccinia virus at 1E6 pfu or VV-TetR-iD13 at 1E7 pfu. Formation of pox lesions on mice feet and tails was monitored (**a**) and quantified (**d**) 5 days post i.v. injection. IVIS imaging was performed on days 5 and 7 post virus injection (**b**) and body weight were measured at consistent intervals (**c**). Organs were harvested at day 10 and tissues processed for quantifying viral titer (**e**). **f, g** Mice were intraperitoneally injected with 1E6 pfu of control and vaccine viruses with serum collected at days 7, 14, and 21. Subsequently, a RBD ELISA assay was performed and endpoint titer for all the samples and days were quantified (**f**). A SARS-CoV-2 pseudotyped neutralization assay was performed on day 14 samples (**g**). **h, i** Mice were intraperitoneally injected with 1E6 pfu of the VV-S-Hexapro-TetR-iD13 in the presence or absence of 625 mg/kg Dox in the diet for two days. Serum was collected at day 14 and was used for an RBD ELISA to quantify the normalized ELISA absorbance versus dilution factor in different conditions (**h**). Serum was also used for a SARS-CoV-2 pseudotyped neutralization assay. **j, k** Mice were intraperitoneally injected with 1E6 pfu of the VV-S-HexaPro-TetR-iD13 or MVA-S-HexaPro. Serum was collected at day 14 and was used for a RBD ELISA to quantify the normalized ELISA absorbance versus dilution factor in different conditions (**j**). Serum was also used for a SARS-CoV-2 pseudotyped neutralization assay (**k**). S and S-HexaPro represent SARS-CoV-2 wild-type spike protein and proline stabilized version of the SARS-CoV-2 spike respectively. Data indicate means ± SD of five (**d**, **f–j**) or ten (**e**) biological replicates. Data indicate means ± SEM of five (**c**) biological replicates. ns $P > 0.05$, *$P < 0.05$ **$P < 0.003841$, ***$P < 0.000125$, ****$P < 0.001$ in unpaired two-samples $t$-test. Source data are provided as a Source Data file.

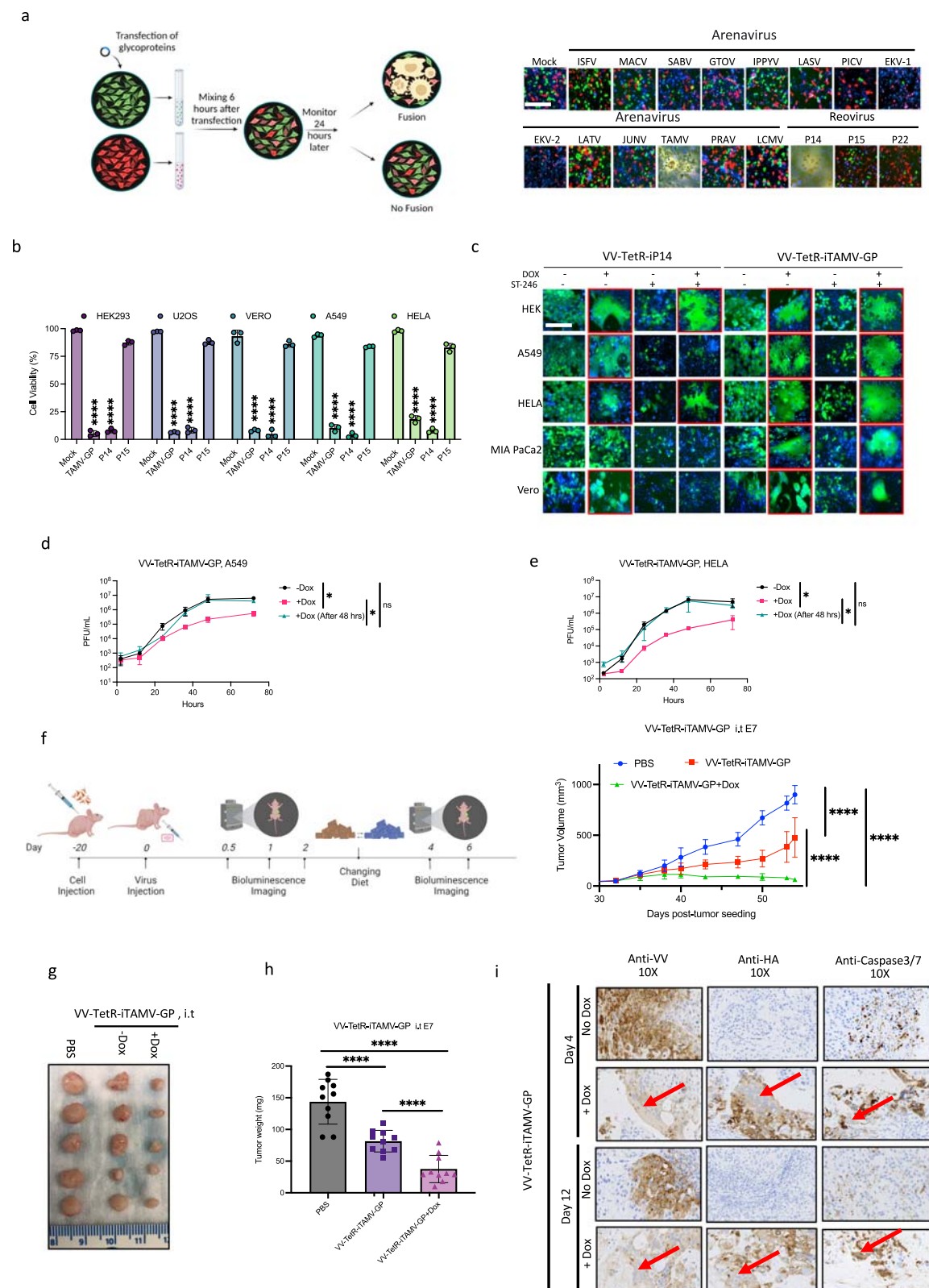

## Discussion

Challenges to the development of replication competent viral vectors as therapeutics include safety concerns related to uncontrolled viral dissemination in cancer patients and healthcare workers as well as off-target delivery of potentially toxic payloads. Here we develop three VV-adapted chemogenetic switches: the rapamycin-inducible ST7 RNA polymerase expression system (FRB/FKBP; Fig. 1), Dox-inducible expression (TetR/TetO; Fig. 2), and cumate-inducible expression (CymR/CuO; Fig. 3) that can be used alone or in combination to control VV replication and/or transgene expression. This study describes, to the best of our knowledge, the first application of the cumate- or rapamycin-inducible expression systems in a replicating viral vector. These systems will facilitate the development of a range of vectors that can potentially deliver potent transgenes in a safe and controlled

**Fig. 7 | Doxycycline-inducible system as an effective strategy to regulate expression of a toxic fusogenic protein. a** Schematic illustration and representative images of GFP expressing cells transfected with Arenavirus glycoproteins including Isfahan virus (ISFV), Machupo virus (MACV), Sabiá virus (SABV), Guanarito virus (GTOV), Ippy virus (IPPYV), Lassa virus (LASV), Pichinde virus (PICV), Ekpoma virus-1 (EKV-1), Ekpoma virus-2 (EKV-2), Latino virus (LATV), Junín virus (JUNV), Tamiami virus (TAMV), Paraná virus (PRAV), lymphocytic choriomeningitis virus (LCMV) and Reovirus P14, P15, P22 FAST proteins. Six hours post transfection of GFP expressing cells, cells were mixed with a separate population of mCherry expressing cells. After 24 h, cells were monitored for signs of cell-cell fusion. **b** HEK293, U2OS, Vero, A549, and Hela cells were transfected with TAMV-GP (Tamiami virus glycoprotein), P14, and P15 FAST proteins. Twenty-four hours after transfection, an Alamar blue viability assay was performed. **c** HEK293, A549, Hela, MiaPaCa2, and Vero cells were infected with either VV-TetR-iP14 or VV-TetR-iTAMV-GP (MOI 0.1) in the presence or absence of 100 ng/ml Dox or ST-246, an anti-poxvirus drug. Representative images were taken 24 h post infection. **d, e** Multistep

growth curves of Hela and U2O2 cells infected with VV-TetR-iTAMV-GP (MOI 0.01), in the presence or absence of 100 ng/ml Dox added at the same time, or 48 h after virus infection. **f–h** A549 tumors were injected with either PBS or VV-TetR-iTAMV-GP (1E7 PFU/tumor) at a size of ~150 mm$^3$. Dox was given in the diet of mice (625 mg/kg) as indicated two days after virus injection. Tumors were measured at regular intervals (**f**). The image shown compares tumor size across all treatment groups and mice in the study (**g**). Tumor weight was also measured (**h**). **i** A549 tumors were injected with VV-TetR-iTAMV-GP (1E7 PFU/tumor) at a size of ~150 mm$^3$ in size and depending on the treatment group received 625 mg/kg of Dox in their diet two days after virus injection. Tumors were harvested at day 4 and day 12 post virus injection and processed for immunohistochemistry staining using vaccinia virus, HA and caspase 3/7 antibodies. Scale bars = 40 μm in (**a, c**). Data indicate means ± SD of three (**b, d, e**) or ten (**f, h**) biological replicates. ns $P > 0.05$, *$P < 0.05$ **$P < 0.003841$, ***$P < 0.000125$, ****$P < 0.001$ in unpaired two-samples $t$-test. Source data are provided as a Source Data file.

fashion to enhance therapeutic outcomes. The use of combinatorial chemogenetic switches is particularly appealing as they provide the clinician with opportunities to tune both the level of virus replication and therapeutic transgene expression. One of the challenges in the development of oncolytic virus vectors is finding the balance between safe selective replication and potency through expression of viral virulence gene products. A strategy that has shown promise is the development of vectors that control certain viral virulence genes using tumor selective promoter elements. For instance oncolytic herpes viruses have been developed for selective replication in glioma cells using the nestin promoter to control the virulence gene ICP 34.5[36]. Our system has the advantage of providing control over the virus replication and gene expression in any tumor type as well as being able to both turn on and off virus gene expression. The cumate-controlled system demonstrated good dynamic range. However, it is not currently an FDA-approved compound thus limiting its rapid application into the clinic. On the other hand, both doxycline and the rapalogs used in this study are FDA-approved drugs and thus their corresponding chemogenetic switches could have a more straight forward regulatory approval path. Ultimately, the expansion of the available toolkits to regulate payload delivery will enable the incorporation of more complex synthetic circuits within viral vectors. Towards this goal, we combined different conditional expression systems (ST7-CymR, ST7-TetR, and Cu-Tet) to create expression systems with two levels of control and decreased basal level of expression. These systems will prove critical to the delivery of potent toxic payloads that can be locally therapeutically effective but have significant toxicity when delivered systemically. Lastly, while our studies have been focused on engineering tunable expression systems in VV, these conditional expression systems can be rapidly adapted to other viral vectors, as we demonstrate for HSV (Fig. 4k–m). We anticipate that each viral vector, with its own unique replication kinetics and replication cycle will have different selectivity and robustness of inducible transgene expression; therefore, expanding the available systems will enable the selection of an expression system tailored to the viral vector or application. Overall, the application of viral expression switches described herein should enable the development of conditionally replicating viral vectors or inducible transgene expression system. This technology will enable testing of highly potent transgenes in oncolytic virus vectors and expand their utility in the clinic.

## Methods

### Ethics
The study discussed here, adheres to all pertinent ethical guidelines at OHRI and the University of Ottawa (certificate for biohazardous material utilization GC317-125-12). The University of Ottawa's institutional animal care committee approved all animal experiments (Protocol ID: OHRI2870 and MEe-2258) which were conducted following

the National Institutes of Health and the Canadian Council on Animal Care standards.

### Viruses
MVA (VR-1508) and VV Copenhagen strain were purchased from the ATCC. The Vaccinia TT strain was a gift from Dr. David Evans. HSV strain KOS was used.

### Cell culture
All cell lines were purchased from the American Type Culture Collections (Manassas, VA). The cells were grown in Dulbecco's Modified Eagle Medium/DMEM (GE Healthcare Life Sciences; ON, CAN) or Roswell Park Memorial Institute/RPMI 1640 Medium (Gibco; MA, USA), supplemented with 10% fetal bovine serum/FBS (Gibco). Cells were kept at 37 °C in a humidified environment with 5% $CO_2$. Cells co-cultured with effector cells were cultivated in RPMI medium supplemented with 10% FBS and 1% penicillin/streptomycin (volume/volume) (Gibco). Co-cultures were kept at 37 °C in a humidified environment with 5% $CO_2$. Cells were regularly checked for mycoplasma contamination using PCR (e-Myo VALiD Detection Kit, 25239, LiliF Diagnostics; South Korea) and were shown to be mycoplasma-free.

### Construct design
The promoter constructs were designed using Snapgene software. codon-optimized inserts were ordered from GenScript (Piscataway, NJ, USA).

### Bioluminescence Imaging
After lysis of virus infected cells with passive lysis buffer, the lysate was moved to 96-well plates and 1:1 ratio of the luciferase reagent (Promega) was added. A Synergy microplate reader (BioTek, Winooski, VT, USA) was used to measure luminescence. Results are presented as RLU normalized to control. The data presented are the mean of three independent experiments.

### In vitro co-culture cell viability assays
The resazurin assay was used to determine cell cytotoxicity caused by viruses or transgenes. The metabolic activity of the cells was measured using the manufacturer's protocol and resazurin sodium salt (R12204; ThermoFisher Scientific). In each well, treated and/or infected cells were given 10% (v/v, final) resazurin and incubated for 2–4 h, depending on the cell line. Using a BioTek Microplate Reader, fluorescence was measured at 590 nm after excitation at 530 nm (BioTek, Winooski, VT, USA). In a nutshell, freshly isolated splenocytes were co-cultured with target viruses in a 24-well plate or a flat-bottom 96-well plate with just media. In short, 24 h before, 200,000 (24-well plate) or 15,000 (96-well plate) target cells were seeded. Virus infections were performed on cells in serum-free media at the indicated MOIs. After 6 h, the media was

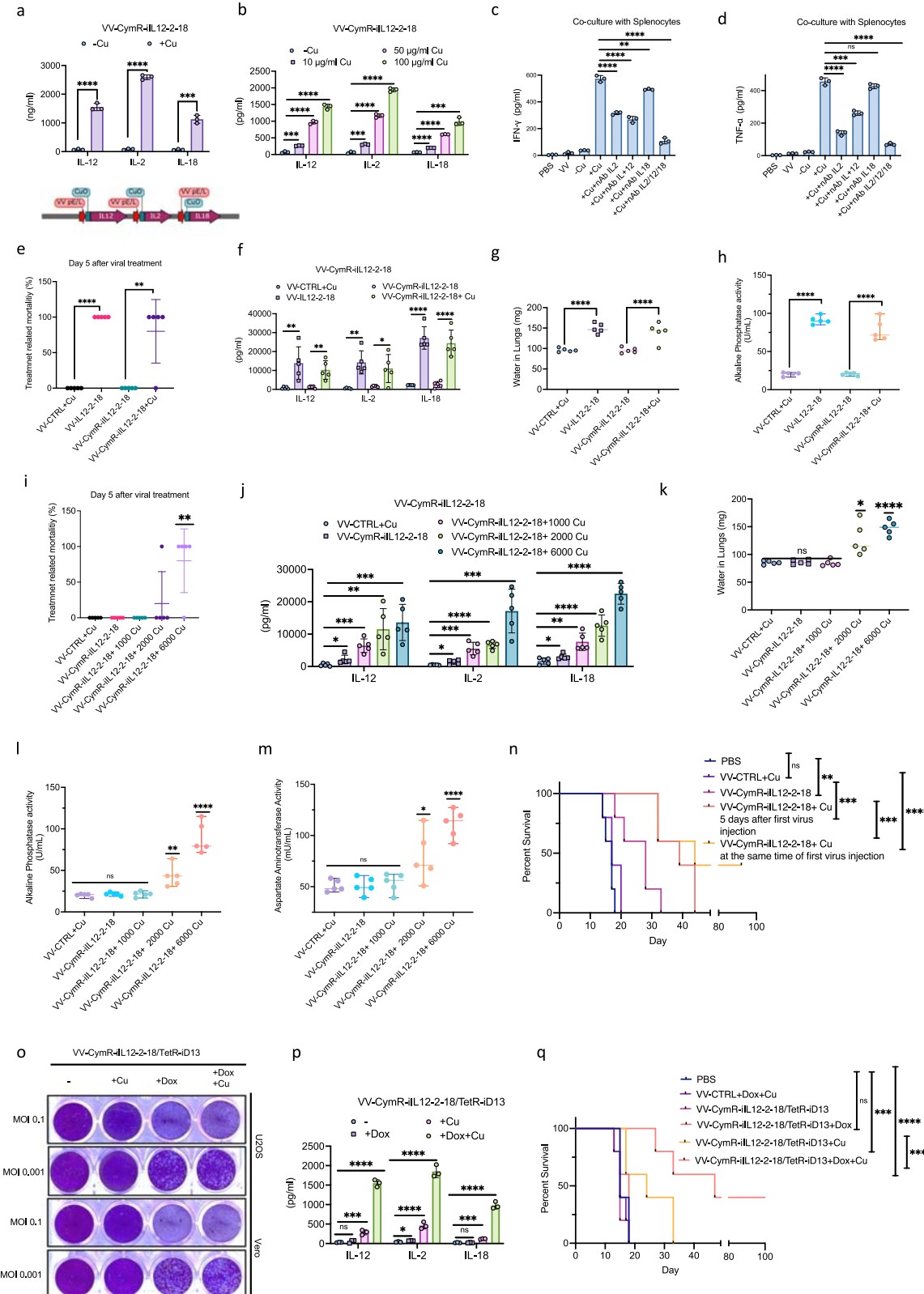

removed and splenocytes in RPMI supplemented with 10% FBS and 1% (v/v) penicillin/streptomycin were added.

## Reagents

Cumate solution (QM150A-1) was purchased from System Biosciences, doxycycline (D9891-25G) was purchased from Sigma-Aldrich, and Rapamycin analogs were purchased from Selleck Chemicals.

## ELISA

Nunc Maxisorp 96-well flat-bottom plates were coated with 125 ng of RBD per well overnight at 4 °C (prepared in-house). The RBD solution was removed the next day, and the plates were washed three times with PBS-Tween (0.1% Tween 20) before being blocked for 1 h with a 3% skim milk solution. Vaccinated mouse sera were then serially diluted in 1% skim milk and added to the plates to incubate for 2 h at room

**Fig. 8 | Applications of the cumate "safety switch" in regulating expression of potentially toxic cytokines. a** IL-2, IL-12 and IL-18 concentrations measured by ELISA from supernatants of U2OS cells 6 h after infection with VV-CymR-iIL12-2-18 (MOI 1) and treated with 100 µg/ml cumate or PBS. **b** IL-2, IL-12 and IL-18 concentrations measured by ELISA from supernatants of U2OS cells 6 h after infection with VV-CymR-iIL12-2-18 (MOI 1) and varying concentrations of cumate. **c, d** IFN-gamma and TNF-alpha levels measured by ELISA. Supernatants were taken from mouse splenocytes cultured ex vivo with conditioned media from U2OS cells treated with 100 µg/ml cumate or PBS following infection with VV-CymR-iIL12-2-18 (MOI 1). Neutralizing antibodies against IL-2, IL-12 and IL-18 were added to conditioned media for 1 h prior to transferring them to splenocytes. **e–h** Treatment-related mortality and toxicity were assessed 5 days after intratumorally injection of HT-29 tumors with control vaccinia virus, VV-IL12-2-18 or VV-CymR-iIL12-2-18 (1E7 PFU/tumor). Tumors were ~150 mm³ in size at the time of injection. Mice treated with VV-CymR-iIL12-2-18 were given a cumate (6000 mg/kg) or regular diet. Water in the lungs (an indication of pulmonary edema) (**g**) and alkaline phosphatase activity (**h**) in serum were measured as indicators of toxicity. **i–m** Treatment-related mortality and toxicity were assessed 5 days after intratumoral injection of HT-29 tumors with control vaccinia virus, VV-IL12-2-18 or VV-CymR-iIL12-2-18 (1E7 PFU/tumor). Tumors were ~150 mm³ in size at the time of injection. Mice treated with VV-

CymR-iIL12-2-18 were given a regular diet, or a diet with varying amounts of cumate (1000, 2000 or 6000 mg/kg; **j**). Water in the lungs (**k**), and serum levels of alkaline phosphatase (**l**) and aspartate aminotransferase activity (**m**) were assessed. **n** Survival analysis of C57BL/6 mice injected intraperitoneally with 5E5 MC38 cells and treated with either PBS, control vaccinia virus, or VV-CymR-iIL12-2-18 at 3E7 pfu/mouse. Treatments were given at days 6, 8, and 10 post cell injection. A cumate diet (1000 mg/kg) was given to the VV-CymR-iIL12-2-18 treatment groups the same day, or 5 days after virus injection as indicated. **o** Crystal violet staining of U2OS and Vero cells 48 h after infection with VV-CymR-iIL12-2-18/TetR-iD13 at different MOIs in the presence of 100 µg/ml cumate (+Cu), 100 ng/ml doxycycline (+Dox), or both (+Dox +Cu). **p** IL-12, IL-2, and IL18 concentrations measured by ELISA in supernatants of U2OS cells 6 h following infection with VV-CymR-iIL12-2-18/TetR-iD13 (MOI 1) in the presence or absence of 100 ng/ml Dox, 100 µg/ml of cumate or both. **q** Survival analysis of C57BL/6 mice injected intraperitoneally with 5E5 MC38-WT cells and treated with either PBS, control vaccinia virus, or VV-CymR-iIL12-2-18/TetR-iD13 at 3E7 pfu/mouse. viruses were injected at days 6, 8, and 10 post cell injection, with mice receiving either Dox (625 mg/kg), cumate (1000 mg/kg) diets, or both for 5 days. Data indicate means ± SD of three (**a–d, p**) or five (**e–n, q**) biological replicates. ns $P > 0.05$, *$P < 0.05$ **$P < 0.003841$, ***$P < 0.000125$, ****$P < 0.001$ in unpaired two-samples $t$-test. Source data are provided as a Source Data file.

---

temperature. In addition, a positive and negative control consisting of a monoclonal RBD antibody (1 g/mL; Cat No: MBS434247, Anti-RBD Domain [SARS-CoV-2 spike], monoclonal antibody, MyBioSource, CA, USA) and a pool of sera taken from mice prior to vaccination were added to each plate. Following the 2-h incubation, plates were washed with PBS-Tween and incubated for 1 h at room temperature with anti-mouse immunoglobulin (Ig)G conjugated to horseradish peroxidase (HRP) (1:3000; Cat No: 314930, Goat anti-Mouse IgG [H + L] Secondary Antibody, HRP, Invitrogen). After that, plates were developed with SigmaFast OPD solution and measured at 490 nm with a BioTek microplate reader.

All experimental absorbance measurements were normalized to the blank and the positive control (monoclonal RBD antibody at 1 g/mL) and fitted with a quadratic binding polynomial assuming 1:1 binding. A Monte Carlo simulation with the nonlinear curve-fitting tool in QtGrace was used to perform the fitting. The reciprocal antibody titer (LDF) was calculated by interpolating the dilution factor that intersected with a minimum detection threshold defined by 10x the standard deviation of responses from TT WT-vaccinated mice, or to a fixed value of 0.025 (whichever was larger).

## Mouse experiments

Six- to 8-week-old female C57BL/6 or CD-1 nude mice (The Jackson Laboratory, Bar Harbor, ME) were obtained for studies. All experiments were approved by the University of Ottawa animal care and veterinary services (MEe-2258-R5, or OHRIe-3340-A), including periodic saphenous vein bleeds for serum collection.

## Pseudovirus neutralization assay

At the time of infection, Vero E6 cells were seeded in 96-well plates with 40,000 cells per well. Serum was first diluted in serum-free DMEM at a 1:10 dilution in a separate 96-well plate, followed by a serial 1 in 2 dilution series. VSV pseudotyped with the SARS-CoV-2 spike glycoprotein and co-encoded with eGFP was then added to serum in an equal volume of serum-free DMEM for a final dilution of 2000 pfu per well and incubated at 37 °C for 1 h. After 1 h, the media on the cell was replaced with 60 L of the virus/serum and incubated at 37 °C for 1 h. The wells were then filled with carboxymethylcellulose (CMC) in DMEM (supplemented with 10% fetal bovine serum) to a final concentration of 3% CMC and incubated at 34 °C for 24 h. Using a Cellomics ArrayScan VTI HCS Reader, GFP foci were imaged and counted.

## Western blot

Whole cell lysates were prepared in Cell Lysis Buffer II (Invitrogen™) with protease/phosphatase inhibitors (NEB). The BCA assay was used

to determine protein concentrations (Pierce). Samples were loaded into precast SDS-PAGE gels after being mixed with NuPage LDS sample buffer (Invitrogen) (Bio-Rad). Proteins were transferred onto nitrocellulose membrane after SDS-PAGE, blocked for 1 h with 5% skim milk in TBS-Tween, and incubated with primary antibody. The following antibodies SARS-CoV-2 Spike (GeneTex, 1A9), HA (Thermo Fisher Scientific, 26183), VV (Abcam, ab117453), and GADPH (Cell Signaling, 2118L) antibodies, which were used at a dilution of 1:1000. Cell Signaling supplied HRP conjugated anti-mouse (7076S) and anti-rabbit (7074S) IgG secondary antibodies, which were used at a dilution of 1:2000. Westerns were created using Bio-Rad Clarity or Clarity Max ECL substrates and imaged with a Bio-Rad GelDoc imaging system. Uncropped western blots are shown in Supplementary Fig. 9.

## Virus production and titration

HeLa cells in 850 cm² roller bottles were infected with virus at a MOI of 0.03 without removing inoculation media to produce vaccinia virus. Cells were cultured at 37 °C in 5% $CO_2$ for 72 h, or until a sufficient cytopathic effect was observed. Cells were pelleted and resuspended in 1 mM Tris at pH 9.0. Cells were frozen/thawed three times before being centrifuged at 700 × $g$ for 10 min at room temperature. Before centrifugation at 20,666 × $g$ for 1 h and 30 min at 4 °C, supernatant was collected and overlaid onto 36% sucrose cushions. Reconstituted viral pellets in 1 ml of 1 mM Tris and stored at −80 °C. Titration was used to determine viral titers. Virus stocks were serially diluted tenfold before infecting U2OS cells in 12-well plates. After 2 h at 37 °C in 5% $CO_2$, the media was replaced with overlay medium (1:1 of 3% CMC and 2x DMEM + FBS) and incubated for 48 h at 37 °C in 5% CO2. Plaques were stained with crystal violet after 48 h and plaque forming units per ml were calculated. Vero cells were used for HSV-1 virus production and titration under the same conditions as described above. Schematic illustration of VV-CTRL, VV-GFPLuc is shown in supplementary Fig. 8.

## Statistical analyses

GraphPad Prism v9 was used to create all graphs and statistical analyses. When appropriate, unpaired two-samples t-test was used. In figure legends, the $n$ value represents all biological replicates. As indicated in the figure legends, error bars represent standard deviation (SD) or standard error of the mean (SEM). For all statistical tests, ns $P > 0.05$, *$P < 0.05$ **$P < 0.003841$, ***$P < 0.000125$, ****$P < 0.001$.

## Reporting summary

Further information on research design is available in the Nature Portfolio Reporting Summary linked to this article.

## Data availability

The authors declare that all data supporting the conclusions of this study are presented within the paper and the supplementary information files and are available from the authors. Schematic pictures for in vivo experiments were created with BioRender.com (Figs. 1i, 2h, 3g, 5g, and 6f). Source data are provided with this paper.

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

## Acknowledgements

We would like to thank other members of the Bell, Diallo, Ilkow, and Auer laboratories. We thank the personnel of the Flow Cytometry Core Facility, Histology Core Facility, and Animal Care and Veterinary Services of the Faculty of Medicine at the University of Ottawa for their support. We thank Dr. Keir Joe Menzies for providing differentiated Myoblast. We thank Bradley Austin, Clarence Choy, Georgia Cheung, Emily Wang, Tina Chen, Emma Le, and Kyle Nguyen for their support and help. Thanks to This Week in Virology for inspiration, specifically Dr. Vincent Racaniello. J.C.B. and C.S.I. support from CIHR, CCSRI and BioCanRx. J.C.B. also supported by Prostate Cancer Canada, and the Terry Fox Research Institute. M.J.F.C., S.B., and T. Azad received funding support from CanPRIME/Mitacs fellowships. M.J.F.C. is funded by the Taggart-Parkes Fellowship. T. Azad is funded by a CIHR postdoctoral award. A.P. was

funded by the Lebovic Fellowship Funding. B.C.D. was supported by National Institute of General Medical Sciences (R35 GM119840) of the National Institutes of Health (NIH).

## Author contributions

J.C.B. proposed and supervised the project. T. Azad, R.R., R.S., S.B., K.A.O., N.T.M., V.H., M.G., M.M., M.J.F.C., H.D.H., A.N.A., M. Ahmadi, N.K.Z. conducted in vitro experiments. T. Azad, R.R., J.P., R.M., X.H. performed mouse experiments. T. Azad, R.R., J.C.B. engineered the viruses. T. Azad, R.R., R.S., A.P. performed data analyses. T. Azad, R.R., R.S. wrote the manuscript. A.G., T. Azad, P.G., T. Alain, M. Ardolina, B.C.D., L.H.T., C.S.I., J.C.B. contributed to the design of studies.

## Competing interests

We declare that J.C.B. has an interest in Turnstone Biologics, which develops the oncolytic Maraba MG1 virus as an OV platform. The remaining authors declare no competing interests.
