## [Peer Review File · Nature Communications]

Reviewers' Comments:

Reviewer #1:

Remarks to the Author:

In this manuscript, the authors Azad et al. engineered a series of gene switches that allow precise control over therapeutically relevant genes delivered by vaccinia virus. Although related gene switches had been explored in other viral vectors already (synthetic circuits for tumor-specific immunomodulator expression delivered by lentiviral vector, see <https://doi.org/10.1016/j.cell.2017.09.049>; synthetic toggle switch to control adenoviral replication and transgenes, see <https://doi.org/10.1038/s41467-019-12794-2>), the current study successfully used orthogonal, US Food and Drug Administration–approved small-molecules as trigger signals to tune transgene expression in oncolytic vaccinia virus. The study provides new means for the construction of genetic circuits, thus adding new tools to improve the safety of oncolytic virus therapies.

However, the authors did not provide adequate in vitro or in vivo data to support their claims, which makes this study less interesting.

1. On page 3, the authors state "Seventy-two hours post-infection and rapalog treatment, the signal dissipated. This was not due to a loss of viral replication in the tumor as rapamycin (i.p.) treatment 48-hour post treatment resulted in marked increase in tumor luciferase signal." The author should provide the detailed data about the duration of transgene expression under a constitutive promoter in the tumor, which may provide more information about the time window for use rapalog in clinics.
2. On page 3 and 4, the authors state "Vaccinia genes have stage-specific transcription factors and corresponding promoter elements which mediate temporal regulation of their expression. We screened a diverse set of VV promoters encompassing the range of temporal expression patterns exhibited by VV proteins." Since vaccinia virus promoter mediate corresponding gene temporal regulation, can authors compare the GFPLuc expression pattern directly transcribed by wild type vaccinia virus promoter with the doxycycline-controlled promoter? In addition, it would be more interesting if the authors explain why use vaccinia virus promoter to construct doxycycline-controlled switch not use some common mammalian promoter.
3. In Supplementary Figure 3L, the bioluminescence activity of VV-GFPLuc is stronger than VV-CymR-iGFPLuc's, which promoter used in VV-GFPLuc? Is the difference caused by the promoter?
4. The authors constructed a series of gene switches by using vaccinia virus promoters. Do authors evaluate the risk of sequence recombination which may be caused by repeating promoter sequences?
5. In Figure 4J, the authors quantified luciferase activity of VV-CymR-TetR-iGFPLuc in HEK293T, Hela, Vero, HT-29, 786-O cells. Does the VV-CymR-TetR-iGFPLuc replicate in HEK293T or other normal cell line? In Figure 5C, Can authors provide the VV-TetR-iD13 and other control virus' performance in normal cell line? Are all the vaccinia virus vectors replicate competent used in each figure?
6. On page 8, "Virus biodistribution in known mouse VV reservoirs were analyzed via plaque assay and there was more a than tenfold decrease in viral titers in lungs, livers, and spleen. Collectively, the data demonstrate that conditionally replicating VV-TetR-iD13 represents a safer vector for antigen delivery." It seems forget to label Figure 6E.
7. On page 8, "Further, we confirmed that VV-S- HexaPro-TetR-D13 induced higher antibody responses relative to a non-replicating vector (MVA) delivering the same antigen (Fig. 6J-K)." It would be more persuasive if the authors use non-replicating vaccinia virus vectors delivering the same antigen as control.
8. In Figure 8N and Figure 8Q, control vaccinia virus, or VV- CymR-iIL12-2-18 at 3E7 pfu/mouse. the treatments were given at days 6, 8, and 10 post cell injection. Why the authors choose this dose regimen? Do the authors try other dose interval, for example given at days 6, 13 and 20 post cell injection?
9. Schematic illustration of VV-CTRL, VV-GFPLuc and other vaccinia virus vectors should be provided in supplementary figures.

Reviewer #2:

Remarks to the Author:

Oncolytic viruses are a potentially effective class of immunotherapeutic agents being tested in otherwise treatment resistant cancers. The incorporation of genetic circuits into oncolytic virus backbones can regulate the timing/magnitude of expression of virulence genes and therapeutic payloads, thereby enabling safer, more effective treatment.

In this manuscript, the team used synthetic virology approaches to develop a new generation of OV's equipped with multiple inducible systems, which can be used as "safety switches" to control both virus replication and transgene expression. This was shown to be precise and effective in controlling virus replication and transgene expression.

The team have shown that this technology can be applied to a variety of inducible systems and is effective in a range of cell lines and in vivo models. The enhanced safety was demonstrated by the ability to limit virus replication in immunodeficient mouse models. For example in the VV-TetR-iD13 infected mice, there was no detected signal, whereas in the VV-CTRL infected mice, there were signals for virus replication in animal tails, paws, and snouts. Furthermore, VV-CTRL infected mice demonstrated increasing weight loss from 4 to 8 days post-infection.

Furthermore, the group showed that control of a virus fusion protein can impact virus propagation. They showed that constitutive expression of the fusion protein impairs viral growth. The use of a chemogenetic switch enabled temporal control of TAMV-GP expression allowing productive virus infection of the tumour before initiating membrane fusion and its downstream therapeutic sequelae.

The potential utility of these technologies are wide and can be used to effectively time oncolytic virus activity in combination with other immunotherapeutics, to increase efficacy, reduce virus clearance and minimise systemic toxicity. This work is therefore very significant and heralds a step-change in oncolytic virus engineering.

The data are presented in a very thorough and scientifically detailed manner, with clear and logical progression in the experiments and figures. The manuscript utilises sound methodology and the standards are very high for the field. There are no errors in the methodology.

Reviewer #1 Comments to Authors

In this manuscript, the authors Azad et al. engineered a series of gene switches that allow precise control over therapeutically relevant genes delivered by vaccinia virus. Although related gene switches had been explored in other viral vectors already (synthetic circuits for tumor-specific immunomodulator expression delivered by lentiviral vector, see <https://doi.org/10.1016/j.cell.2017.09.049>; synthetic toggle switch to control adenoviral replication and transgenes, see <https://doi.org/10.1038/s41467-019-12794-2>), the current study successfully used orthogonal, US Food and Drug Administration–approved small-molecules as trigger signals to tune transgene expression in oncolytic vaccinia virus. The study provides new means for the construction of genetic circuits, thus adding new tools to improve the safety of oncolytic virus therapies. However, the authors did not provide adequate *in vitro* or *in vivo* data to support their claims, which makes this study less interesting.

We appreciate the reviewer acknowledging the significance of our research in enhancing the safety and efficacy of oncolytic virus therapies. In the following we describe multiple *in vitro* and *in vivo* experiments providing more data that support our interpretation of the data and address each of this reviewer’s comments.

1. On page 3, the authors state “Seventy-two hours post-infection and rapalog treatment, the signal dissipated. This was not due to a loss of viral replication in the tumor as rapamycin (*i.p.*)

treatment 48-hour post treatment resulted in marked increase in tumor luciferase signal.” The author should provide the detailed data about the duration of transgene expression under a constitutive promoter in the tumor, which may provide more information about the time window for use rapalog in clinics.

Upon reflection we realized that we did not make it clear exactly how this experiment was conducted which has led to some confusion. In this study, we infected tumours and applied a single dose of the rapalog at the time of infection using the clinically appropriate route for each rapalog. As shown in **Fig. 1**, we observed robust stimulation of luciferase 24 hours later. The animals received no further treatments with the rapalogs and then at 72 hours post initial infection the tumours were once again imaged to determine the level of luciferase expression. As is clear in **Fig 1**, the level of expression of luciferase driven by the initial induction with the raplog has decreased significantly by 72 hours after drug administration. This decrease in signal is not related to clearance of virus as we determined the level of infectious particles at day 10 and it was comparable in all of the tumours sampled. This additional new data is included in **Supplementary Figure 1d** and below. Furthermore, in animals that were infected but not treated with rapalog there was minimal luciferase signal at 24 hours which was robustly induced upon rapalog induction at 48 hours. We have modified the text in the body of the manuscript (see page 3, lines 37-45) to provide this clarification. As the referee suggests this initial data does suggest some time frames for clinical testing and certainly future studies will prepare a more fulsome data package to support clinical translation.

2. On page 3 and 4, the authors state “Vaccinia genes have stage-specific transcription factors and corresponding promoter elements which mediate temporal regulation of their expression. We screened a diverse set of VV promoters encompassing the range of temporal expression patterns exhibited by VV proteins.” Since vaccinia virus promoter mediate corresponding gene temporal regulation, can authors compare the GFP_{Luc} expression pattern directly transcribed by wild type vaccinia virus promoter with the doxycycline-controlled promoter?

We conducted a comparison of luciferase expression at various time points using a wild-type vaccinia virus promoter (P11) and the doxycycline-controlled promoter with and without induction. Our findings show that there is a comparable level of reporter expression between the wild-type P11 vaccinia virus promoter and the doxycycline-controlled P11 promoter when Dox is added. This new data has been added to **Supplementary Figure 2c** and below.

In addition, it would be more interesting if the authors explain why use vaccinia virus promoter to construct doxycycline-controlled switch not use some common mammalian promoter.

Overall, while vaccinia virus is able to infect and replicate in mammalian cells, its transcriptional machinery and genomic structure is not optimized for the use of mammalian promoters as reported in the previous publications (Grimm et al., Nature Structural & Molecular Biology 2021, <https://doi.org/10.1038/s41594-021-00655-w>). Vaccinia virus has its own set of regulatory sequences and transcription factors that are adapted to its specific needs, and the use of mammalian promoters is not compatible with these requirements. Vaccinia virus replication occurs in the cytoplasm and thus does not have access to the nuclear transcriptional machinery. To unequivocally test this hypothesis we created a virus with a CMV promoter driving luciferase in vaccinia virus. As expected, we did not observe any activity of this promoter in vaccinia virus. This new data has been added to **Supplementary Figure 1a**.

3. In Supplementary Figure 3L, the bioluminescence activity of VV-GFPLuc is stronger than VV-CymR-iGFPLuc's, which promoter used in VV-GFPLuc? Is the difference caused by the promoter?

Thank you for the excellent questions. We utilized different promoters for VV-GFPLuc (p7.5) and VV-CymR-iGFPLuc (p11), which we acknowledge may not be the most appropriate control. We have now updated Supplementary Figure 3L to have the same promoter in VV-GFPLuc and VV-CymR-iGFPLuc (p11 in both). Even with the same promoter, we observed a significantly lower luciferase signal from VV-CymR-iGFPLuc compared to VV-GFPLuc. To investigate which promoter is more suitable for achieving higher expression levels, we tested several promoters and found that PEL yields a significantly stronger signal when used in cumate chemogenetic systems (**Figure 3b**). As we had emphasized in the manuscript, this highlights the importance of selecting the optimal viral promoter for each chemogenetic switch to achieve the

best possible dynamic range. This is another important and unexpected innovation of our study and we have now reinforced this in the text.

4. The authors constructed a series of gene switches by using vaccinia virus promoters. Do authors evaluate the risk of sequence recombination which may be caused by repeating promoter sequences?

This is an excellent point and a valid concern. We have deemed the overall risk of recombination low as:

a) the introduced vaccinia promoters are significantly smaller (20-100 base pairs) compared to other mammalian promoters like CMV and EF1-a (1000-1500 base pairs), which reduces the likelihood of recombination.

b) the vaccinia virus is a cytoplasmic virus that is well-known for its stability and low rate of recombination, making it an excellent platform as a replicating gene therapy vector. As an example, we have been using vaccinia virus expressing T7 for over 15 years and after sequencing, we have not detected any mutation in our virus or its transgene.

c) As part of our quality control, we regularly confirm the stability of the viral sequences, using Nanopore sequencing. We have, to-date, not observed any sequence recombination in the vectors containing chemogenetic switches after numerous rounds of passaging.

5. In Figure 4J, the authors quantified luciferase activity of VV-CymR-TetR-iGFP Luc in HEK293T, Hela, Vero, HT-29, 786-O cells. Does the VV-CymR-TetR-iGFP Luc replicate in HEK293T or other normal cell line? In Figure 5C, Can authors provide the VV-TetR-iD13 and other control virus' performance in normal cell line? Are all the vaccinia virus vectors replicate competent used in each figure?

The vaccinia virus and all other viruses we utilized were shown to replicate in HEK293, HeLa, U2OS, Vero, and A549 cells, as demonstrated in Fig 1D, 2D, 6E-F, 8F-G and Supplementary Figure 1A-B, 2B-C, 3E-G. Since we are using an oncolytic virus with a deletion in the TK genes, its replication is significantly impaired in normal cells. The properties of this attenuation have been established in previous work (Evgin et al., Immunobiology 2012, <https://doi.org/10.1016/j.imbio.2012.08.018>; Crupi et al., Frontiers in Immunology 2022, <https://doi.org/10.3389/fimmu.2022.1029269>; Parato et al., Molecular Therapy 2012, <https://doi.org/10.1038/mt.2011.276>). In addition, we confirmed the selectivity in cancer cells by screening our virus in HEK293T, HELA cells, as well as several normal cell types. We observed these viruses replicates 100-1000-fold less in human primary cells. This additional data is now reported in **Supplementary Fig 4d**.

6. On page 8, “Virus biodistribution in known mouse VV reservoirs were analyzed via plaque assay and there was more a than tenfold decrease in viral titers in lungs, livers, and spleen. Collectively, the data demonstrate that conditionally replicating VV-TetR-iD13 represents a safer vector for antigen delivery.” It seems forget to label Figure 6E.

We have double-checked all the figures and the manuscript to ensure all figures are properly labelled and cited within the text.

7. On page 8, “Further, we confirmed that VV-S- HexaPro-TetR-D13 induced higher antibody responses relative to a non-replicating vector (MVA) delivering the same antigen (Fig. 6J-K).” It would be more persuasive if the authors use non-replicating vaccinia virus vectors delivering the same antigen as control.

We apologize to the review for the lack of clarity on the vaccinia virus vectors used in this study. We have now revised the manuscript to emphasize that MVA is a non-replicating vaccinia vector (see page 8, lines 32-34); hence in **Figure 7 J-K**, it serves as the non-replicating control vaccinia virus vector to deliver the same antigen.

8. In Figure 8N and Figure 8Q, control vaccinia virus, or VV- CymR-iIL12-2-18 at 3E7 pfu/mouse. the treatments were given at days 6, 8, and 10 post cell injection. Why the authors choose this dose regimen? Do the authors try other dose interval, for example given at days 6, 13 and 20 post cell injection?

We selected these time points based off our previous work examining OV efficacy in these mouse models (Whelan et al., *Frontiers in Immunology* 2023, <https://doi.org/10.3389/fimmu.2022.1050250>, Wedge et al., *Nature Communications* 2022, <https://doi.org/10.1038/s41467-022-29526-8>). These time points are commonly used in the OV field to evaluate oncolytic virotherapy (Zuo et al., *eBioMedicine* 2021, <https://doi.org/10.1016/j.ebiom.2021.103240>; Liu et al., *Nature Communications* 2018, <https://doi.org/10.1038/s41467-018-06954-z>). While we acknowledge the reviewer’s point that timing of dosing can influence therapeutic outcome of oncolytic virotherapy, the major focus of this study is to illustrate the temporal control and potential therapeutic benefit provided by incorporation of these chemogenetic switches into viral vectors. Our results serve as a proof of concept illustrating the value of these on/off switches for controlling toxicity and increasing

efficacy. Defining the optimal timing of induction is beyond the scope of this study; however, we are actively exploring the kinetics for chemical induction, and this will be the subject of an independent manuscript.

9. Schematic illustration of VV-CTRL, VV-GFP_{Luc} and other vaccinia virus vectors should be provided in supplementary figures.

This is an excellent suggestion that will help readers better understand the experimental setup. We have added Schematic illustration of VV-CTRL, and VV-GFP_{Luc} to **Supplementary Figure 8**.

Reviewer #2 (Remarks to the Author):

Oncolytic viruses are a potentially effective class of immunotherapeutic agents being tested in otherwise treatment resistant cancers. The incorporation of genetic circuits into oncolytic virus backbones can regulate the timing/magnitude of expression of virulence genes and therapeutic payloads, thereby enabling safer, more effective treatment.

In this manuscript, the team used synthetic virology approaches to develop a new generation of OV's equipped with multiple inducible systems, which can be used as "safety switches" to control both virus replication and transgene expression. This was shown to be precise and effective in controlling virus replication and transgene expression.

The team have shown that this technology can be applied to a variety of inducible systems and is effective in a range of cell lines and in vivo models. The enhanced safety was demonstrated by the ability to limit virus replication in immunodeficient mouse models. For example in the VV-TetR-iD13 infected mice, there was no detected signal, whereas in the VV-CTRL infected mice, there were signals for virus replication in animal tails, paws, and snouts. Furthermore, VV-CTRL infected mice demonstrated increasing weight loss from 4 to 8 days post-infection.

Furthermore, the group showed that control of a virus fusion protein can impact virus propagation. They showed that constitutive expression of the fusion protein impairs viral growth. The use of a chemogenetic switch enabled temporal control of TAMV-GP expression allowing productive virus infection of the tumour before initiating membrane fusion and its downstream therapeutic sequelae.

The potential utility of these technologies are wide and can be used to effectively time oncolytic virus activity in combination with other immunotherapeutics, to increase efficacy, reduce virus clearance and minimise systemic toxicity. This work is therefore very significant and heralds a step-change in oncolytic virus engineering.

The data are presented in a very thorough and scientifically detailed manner, with clear and logical progression in the experiments and figures. The manuscript utilises sound methodology and the standards are very high for the field. There are no errors in the methodology.

Thanks to the reviewer for the kind comment and recognizing the significance of our work.

Reviewers' Comments:

Reviewer #1:

Remarks to the Author:

I have no more comments.